# In vivo screen reveals specific roles of Hippo pathway components in development and regeneration

Susanna E Riley[1], Michaela Noskova Fairley[1], Shijia Xia[1], Richard Cunningham[1], Justyna Cholewa-Waclaw[2], Yi Feng[1,3], Carsten Gram Hansen[1]

The Hippo signalling pathway is a major regulator of regeneration and development. However, the comparative importance and functional roles of individual Hippo pathway components in vivo are greatly unknown, particularly within the vertebrate lineage. To gain direct and comparable insights, we took advantage of the zebrafish larva model system. We generated individual and combined CRISPR/Cas9 F0 knockouts of a range of core Hippo pathway genes, including upstream regulators, the co-transcriptional regulators Yap1/Taz, and Yap1/Taz target genes. We analysed and compared the resulting developmental and regenerative phenotypes. Our findings highlight that paralogues of core components have distinct, but in some instances overlapping, functions. Intriguingly, we find that Yap1 and Taz have differential roles during development and regeneration. In addition, we characterise and compare two tail fin regenerative paradigms: after both severe and mild injury. These injury paradigms are drastically different and elicit diverse resolution processes. We confirm critical roles of the immune system in the regenerative process. Macrophage recruitment is reduced during severe tail fin regeneration after Yap1 and Taz loss, appearing earlier in *yap1* than *wwtr1* Crispants and correlating with defective regenerative function. This defective macrophage involvement might therefore be one of the mediators of the deficient regeneration in these two Crispants. Overall, our analysis emphasises distinct requirements and responses of the Hippo pathway during development and across different regenerative paradigms.

## Introduction

During development and regeneration, tissues undergo dramatic changes. Cells integrate positional and intrinsic cues, leading to cell state transitions, migration, and proliferation. These dynamic events must be precisely integrated spatio-temporally to ensure coordinated processes (1, 2, 3). One signalling pathway implicated in both development and regenerative processes across species is the Hippo pathway (4, 5, 6, 7, 8, 9, 10). However, the relative importance and role of individual Hippo pathway components in these complex biological processes in vertebrates are not fully elucidated (1). The Hippo pathway is a multi-component complex signalling cascade composed of an upstream inhibitory serine/threonine kinase module, which when activated phosphorylates and inhibits the co-transcriptional regulators YAP and TAZ (Fig 1A). The proximal kinases directly phosphorylating YAP and TAZ are the LATS1/2 kinases (11, 12, 13, 14). When YAP and TAZ are not phosphorylated on a specific set of serine residues, these co-transcriptional regulators preferentially localise to the nucleus (14, 15, 16, 17), where they mediate transcription including of the target genes *CYR61* (*CCN1*) and *CTGF* (*CCN2*) (14, 17, 18, 19, 20). Here, we set out to establish a comparative analysis of the roles of core components within the Hippo pathway (Fig 1A).

Seminal work carried out through mosaic screens in the fruit fly (21, 22, 23, 24, 25, 26, 27, 28, 29) combined with the generation of knockout libraries in mammalian tissue culture cells (13, 30, 31, 32, 33) has been transformative in identifying the functional relevance and detailing mechanistic insights into the role of individual Hippo pathway components. Indeed, the pathway was named after the Hippo-like overgrowth phenotype observed in mosaic screens of the fruit fly, in which the Hpo kinase (MST1/2 in humans) was identified (26, 27, 28, 29). One advantage of the fruit fly is that, in general, only one gene encodes each core component, whereas in the vertebrate lineage, usually two paralogues exist (34), including YAP/TAZ (Yki), MST1/2 (Hpo), and LATS1/2 (Wts) (34, 35). This limited redundancy made the initial identification of fundamental components more straightforward but does not fully reflect the complexities of the vertebrate lineage (34, 35).

So far, no side-by-side comparison of Hippo pathway components has been carried out in vertebrate animals. Taz (*Wwtr1*)$^{-/-}$ mice are viable but exhibit various severe phenotypes, including polycystic kidney and pulmonary disease (36, 37, 38), whereas Yap1 is essential for development in mice. Yap1 deficiency is embryonic lethal, with *Yap1*$^{-/-}$ mice dying around E8.5 because of severe

[1]Centre for Inflammation Research, Institute for Regeneration and Repair, Edinburgh BioQuarter, University of Edinburgh, Edinburgh, UK [2]Institute for Regeneration and Repair, The University of Edinburgh, Edinburgh BioQuarter, Edinburgh, UK [3]Cancer Research UK Scotland Centre, Institute of Genetics and Cancer, University of Edinburgh, Edinburgh, UK

Correspondence: Carsten.G.Hansen@ed.ac.uk

# Crispant generation – Yap and Taz (*wwtr1*)

**A**

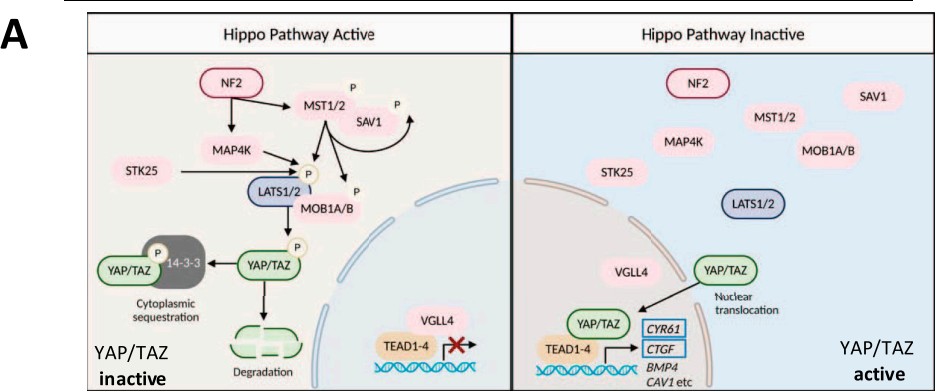

**Figure 1. Generation of Yap1 and Taz (*wwtr1*) Crispants.**
**(A)** Schematic of the core Hippo pathway, consisting of an upstream serine/threonine kinase cascade, which when active causes LATS1/2 activation, and thereby phosphorylation and inhibition of the co-transcriptional mediators YAP and TAZ. When the Hippo pathway is on, YAP and TAZ locate to the cytoplasm (left). When LATS1/2 are inactive (Hippo off, right), YAP and TAZ localise to the nucleus where they bind to cognate transcription factors. Components targeted in the CRISPR screen are outlined, as well as the target genes *CYR61*, also known as *CCN1*, and *CTGF*, also known as *CCN2*. **(B)** Western blot probed for Yap1 in *yap1* Crispants and controls. Note the absence of Yap1 is only apparent in Crispants. **(C)** Western blot probed for Taz in *wwtr1* Crispants and controls. Note the absence of Taz protein in *wwtr1* Crispants only. **(D)** Western blot probed for Taz in *yap1* Crispants and controls. No change is observed. **(E)** Western blot probed for Yap1 in *wwtr1* Crispants and controls. There is no apparent change in Taz levels. (B, C, D, E) 72 hpf Crispants, Ponceau shown to ensure equal loading. Controls are WT, Cas9 protein injected only (Cas9), and guide RNA only (gRNA). **(F)** Western blot quantification of *yap1* Crispant samples. *yap1* Crispants have reduced Yap1, but not Taz. **(G)** Western blot quantification of *wwtr1* Crispant samples reveals Taz loss, but not Yap1. (F, G) Data shown are from three independent repeats; each dot represents one repeat. Bars show the mean ± SD. Significance is calculated by a one-sample *t* test. **P < 0.01; ***P < 0.001; ns, not significant.

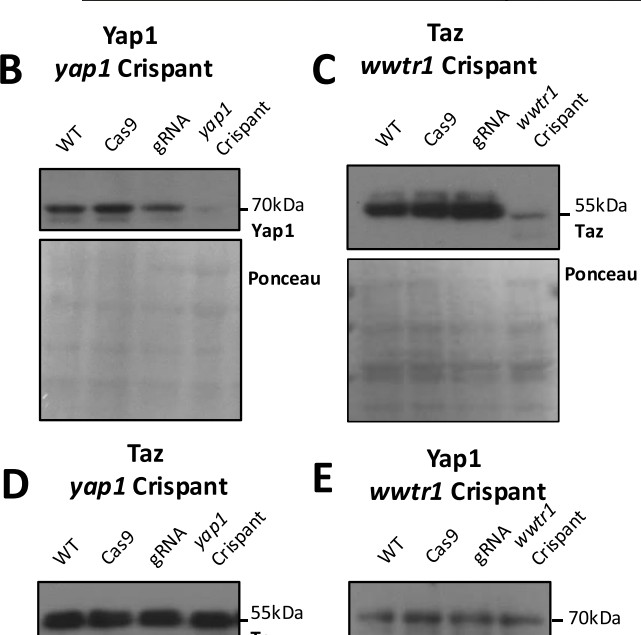

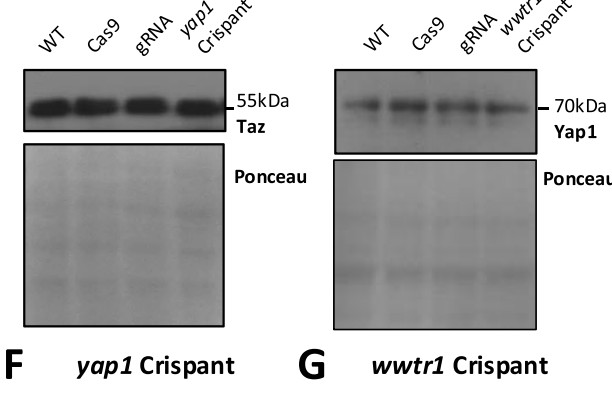

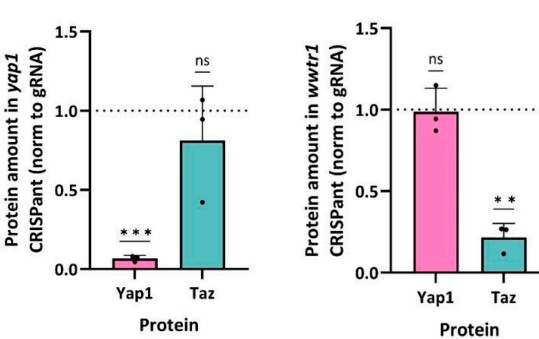

deformities including failure in yolk sac vasculogenesis, chorio-allantoic attachment, and embryonic axis elongation (39). This makes studying the effect of global Yap1 deficiency within mice during development and regeneration impossible, precluding a full mechanistic dissection of events during development. Dual-knockout (KO) $Yap1^{-/-};Wwtr1^{-/-}$ embryos die before the morula stage (16–32 cells), before establishment of inside and outside cell populations (10). This indicates that Yap1 and Taz are essential but play redundant roles in morula formation (10, 40).

Zebrafish have a substantial regenerative capacity in most body parts (1, 41, 42). The optical transparency of zebrafish larvae, combined with fast development from fertilised eggs to free swimming larvae (5 d) and the ease of efficient CRISPR/Cas9-mediated F0 knockout (1, 41, 42), allows for imaging-based phenotypic analysis and rapid genetic screens. The Crispant F0 knockout approach has been shown to be powerful in obtaining insights into multiple processes, including cardiac development and function (43), and control of inflammation after spinal cord injury (44).

Prior studies have highlighted various Hippo pathway components in zebrafish development, physiology, and regenerative processes (1). These studies demonstrate general developmental defects in Hippo pathway mutants including reduced survival in *yap1*- and *wwtr1*-disrupted embryos, as well as the upstream kinase cascade component *sav1*-, *stk3*-, and *lats2*-disrupted embryos (45, 46, 47, 48) and developmental delay in *yap1*-, *wwtr1*-, *amotl2*-, *lats1*-, and *lats2*-disrupted embryos (47, 48, 49, 50, 51) (Fig 1A). In addition, defects in organogenesis, such as incorrect heart looping, are present in *yap1*- and *wwtr1*-disrupted embryos (52). The Hippo pathway is integrated into multiple regenerative processes, with data indicating that an increase in Yap1 and Ctgfa activity promotes regeneration in a range of contexts, including during tail fin and lateral line regeneration (1, 53, 54, 55, 56, 57, 58). These studies underscore conserved developmental and central regenerative roles of the Hippo pathway in the zebrafish (1).

Collectively, past studies used multiple complementary approaches, including morpholinos and the generation of stable mutant lines. Each approach has their strengths and limitations (59, 60), including the inability of generating a range of stable mutants for screen purposes within a feasible workflow. In addition, first-generation Yap1/Taz-Tead inhibitors (61, 62, 63) have been used. Further inhibitors (64), including LATS inhibitors, are being developed (65, 66); however, these are not yet fully characterised. Hence, the comparative and specific compound effect on paralogues and the conservation of use in zebrafish orthologues are not evaluated. In addition, there is a general lack of drugs targeting non-enzymatic proteins within the pathway. These prior approaches therefore do not allow for direct comparative analysis of complete loss of activity across multiple Hippo pathway components in vivo. Here, we take advantage of the efficient CRISPR/Cas9-mediated F0 knockout approach in zebrafish larvae (1, 41, 42, 67, 68, 69) as a unique model system allowing us to directly compare the role of a core set of Hippo pathway components in development and regeneration (Fig 1A). We targeted fish orthologues of the upstream inhibitory kinase cascade components NF2, LATS1, and LATS2 (13, 16, 30, 70, 71, 72), the transcriptional co-activators YAP and TAZ (encoded by *WWTR1*), and the YAP/TAZ

target genes *CYR61* (also known as *CCN1*) and *CTGF* (*CCN2*) (17, 33, 73) (Fig 1A). Notably, a near-complete gene duplication event took place in the teleost; consequently, zebrafish have multiple copies of some genes (1, 74). In these instances, we targeted both the "a" and "b" paralogues.

Overall, our selection of core components of the Hippo pathway encompasses a range of proteins at different levels within this signalling cascade, with mutations expected to display diverse effects (1) (Fig 1A). Our analysis provides insights into the fundamental roles these components play in development, organ size control, and regeneration within the vertebrate lineage (75).

# Results

## Two-guide Crispants result in near-complete gene disruption

In order to use F0 Crispant zebrafish larvae for our targeted Hippo pathway functional screen, we sought to establish the efficiency of our CRISPR/Cas9-mediated knockout approach. We designed CRISPR/Cas9 guide sequences targeting a range of core components of the Hippo pathway (Fig 1A) and injected them into zebrafish embryos (two guides per component) at the single-cell stage along with the Cas9 protein. The efficiency of the two guides and the resulting Crispant mutagenesis was analysed by restriction enzyme digest of the larval DNA (Fig S1). These restriction analyses highlight that our protocol is highly efficient, with well above 90% mutagenesis in individual larvae and conserved across genes (Fig S1). To establish that the CRISPR/Cas9 mutagenesis also causes loss of protein, we carried out Western blots (WBs) on lysates from pooled 72 hours post-fertilisation (hpf) larvae and probed for the co-transcriptional regulators Yap1 (Fig 1B) and Taz (encoded by the gene *wwtr1*) (Fig 1C) in their respective Crispants. These WBs reveal an almost complete loss of Yap1 protein in *yap1* Crispants and Taz protein in *wwtr1* Crispants, confirming that the approach allows for individual Crispant larvae to be analysed with confidence. As compensatory mechanisms and functional interdependencies have been reported between YAP and TAZ (10, 34, 38, 76, 77, 78), we examined the protein levels of Taz in *yap1* Crispants (Fig 1D and F), and Yap1 in *wwtr1* Crispants (Fig 1E and G). This reveals no substantial compensatory response or interdependency (Fig 1D–G).

## The Crispant screen reveals redundant and specific developmental Hippo component roles

After validating our gene targeting approach, we next sought to establish the functional importance that individual Hippo pathway components have on development. We targeted selected core components (Fig 1A) in the developing zebrafish and analysed their gross phenotypes at 96 hpf, including length, swim bladder, and eye area (Fig 2A and B). Where present within our selected genes, we targeted both components with fish-specific gene duplications (*nf2a/nf2b* and *ctgfa/ctgfb*) (1, 74), and in addition targeted closely related components (*yap1/wwtr1* and *lats1/lats2*) (Fig 2).

Notably, our data show that individual upstream components have distinct phenotypes, including decreased larva length in *yap1*

# Development Phenotypes

## A    96hpf representative images - Crispants

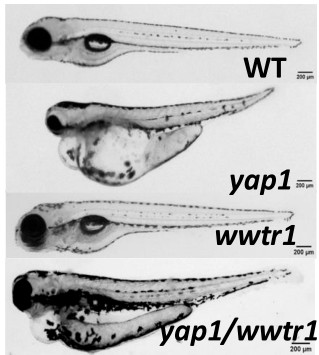
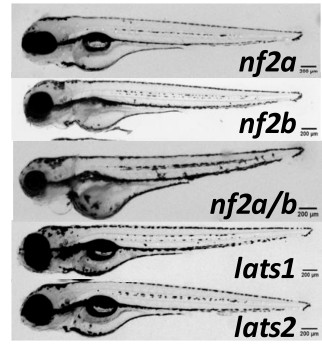
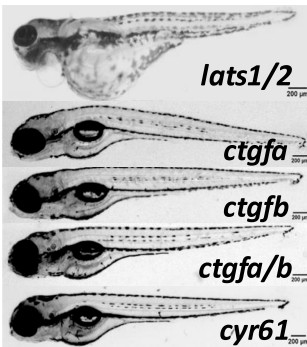

## B    Parameters measured

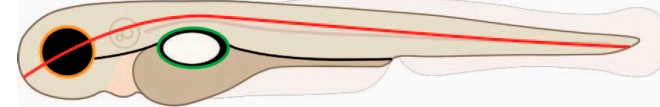

Length
Swim bladder
Eye

## Development Phenotypes - Quantification

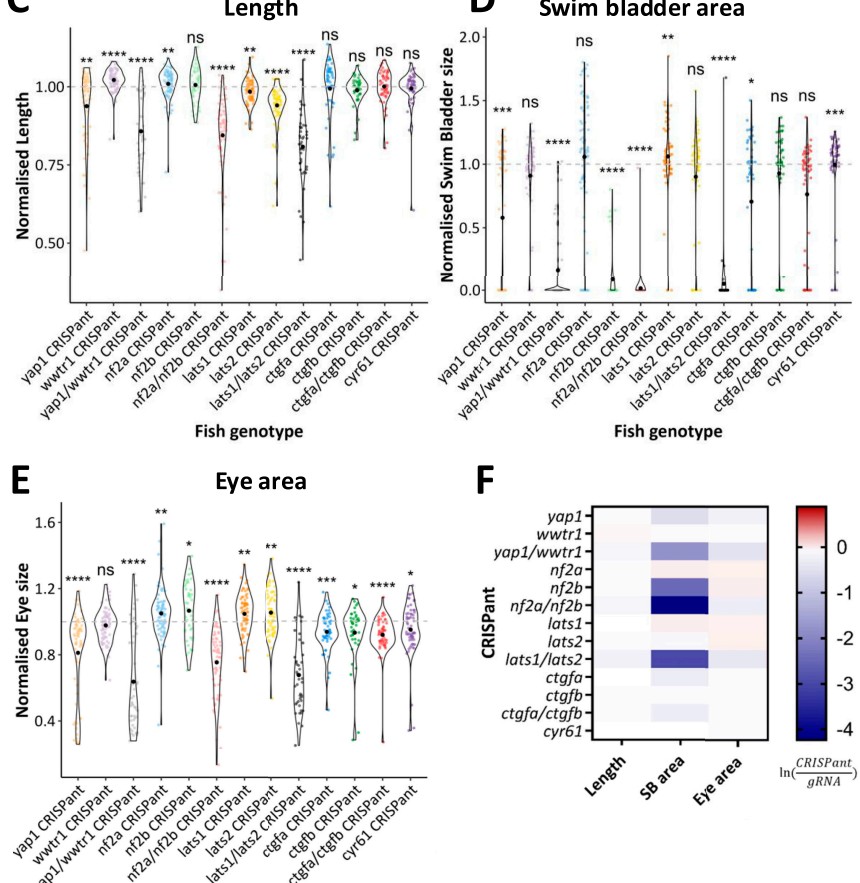

**Figure 2.  Development phenotypes in Crispants.**

**(A)** Representative images of 96 hours post-fertilisation (hpf) WT (top left) and Crispants. Scale bar = 200 μm. **(B)** Schematic of parameters measured at 96 hpf. **(C)** Quantification of embryo length in Crispants, normalised to gRNA controls. Embryo length both increased and decreased after CRISPR mutagenesis. **(D)** Quantification of swim bladder area (as a percentage of body area) in Crispants, normalised to gRNA controls. The swim bladder size decreased following the majority of CRISPR mutagenesis. **(E)** Quantification of eye area (as a percentage of body area) in Crispants, normalised to gRNA controls. Note the eye area is increased after mutagenesis of individual upstream Hippo pathway kinases but decreased in double Crispants and in many downstream components, notably except in *wwtr1* Crispants. **(F)** Heat map of average phenotypes. Values were calculated by taking the natural log (ln) of the average normalised value for each Crispant group. Positive value (red) denotes an increase from controls, whereas a negative value (blue) denotes a reduction from controls. Significance in (C, D, E) is calculated by the Wilcoxon test (μ = 1). Data are from at least three independent experimental repeats. Each dot represents one embryo. Values (C, D, E) are normalised to the average value for gRNA controls from the same independent repeat. *P < 0.05; **P < 0.01; ***P < 0.001; ****P < 0.0001; ns, not significant.

# Tail fin regeneration – Crispants

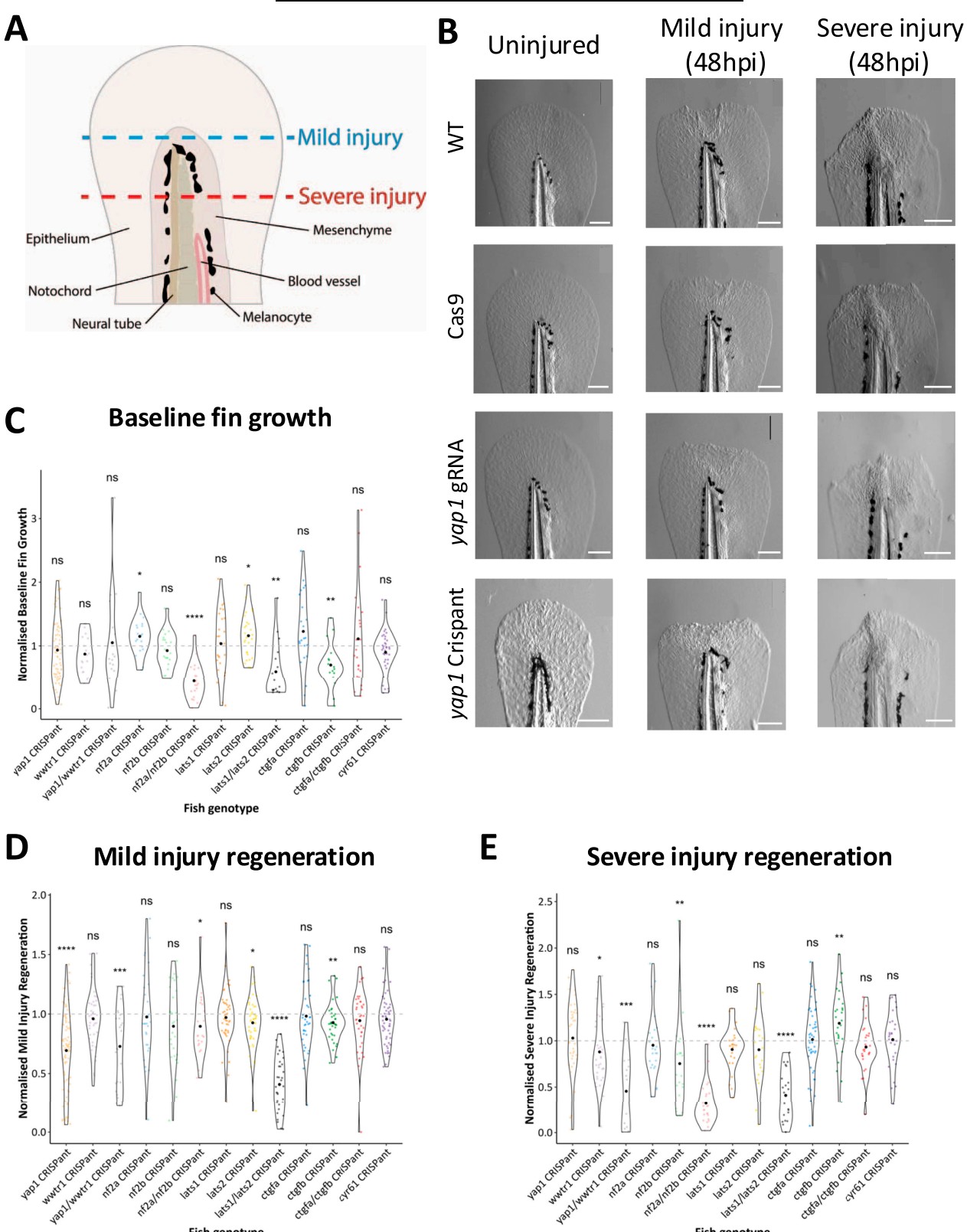

**A**

Epithelium
Notochord
Neural tube
Mesenchyme
Blood vessel
Melanocyte
- Mild injury
- Severe injury

**B**

|  | Uninjured | Mild injury (48hpi) | Severe injury (48hpi) |
|---|---|---|---|
| WT | | | |
| Cas9 | | | |
| *yap1* gRNA | | | |
| *yap1* Crispant | | | |

**C** Baseline fin growth

Normalised Baseline Fin Growth vs Fish genotype

**D** Mild injury regeneration

Normalised Mild Injury Regeneration vs Fish genotype

**E** Severe injury regeneration

Normalised Severe Injury Regeneration vs Fish genotype

**Figure 3. Crispant tail fin regeneration.**
**(A)** Schematic of injury points of a 3 dpf embryo tail fin. Red dotted line = severe injury amputation plane; blue dotted line = mild injury amputation plane.
**(B)** Representative images of uninjured growth, and mild and severe tail fin regeneration in *yap1* Crispants and controls. Cas9 = Cas9-only–injected fish; gRNA = *yap1* CRISPR guide RNA-only–injected fish; Crispant = *yap1* CRISPR-mutagenised fish. Scale bar represents 100 *µ*m. **(C)** Quantification of uninjured fin growth in Crispants, normalised to gRNA controls. Baseline (uninjured) fin growth was altered in some Crispants, most severely in upstream component double Crispants. **(D)** Quantification of regeneration after mild tail fin injury in Crispants. **(E)** Quantification of regeneration after severe tail fin injury in Crispants. Regeneration is reduced in many Crispants but notably increased in *ctgfb* Crispants. Note the reduction in regeneration in *wwtr1* Crispants, the only *wwtr1*-specific regeneration phenotype observed throughout the screen. Significance is calculated by the Wilcoxon test ($\mu$ = 1). **(C, D, E)** Data are from at least three independent experimental repeats. Each dot represents one embryo. Values are normalised to the average value for gRNA-only controls from the same independent repeat. *$P < 0.05$; **$P < 0.01$; ***$P < 0.001$; ****$P < 0.0001$; ns, not significant.

Crispants, but increased larva length in *wwtr1* Crispants (Fig 2C), and with clear differential effects when targeting one paralogue compared with the other (Fig 2C–E). These differences across paralogues also include decreased swim bladder and eye area in *yap1* Crispants that are not phenocopied in *wwtr1* Crispants (Fig 2D and E). In addition, *nf2a* Crispants are longer (Fig 2C) and have increased eye size (Fig 2E) compared with controls, phenotypes that are not copied in *nf2b* Crispants. This directly shows a functional separation between multiple paralogues within the Hippo pathway, highlighting the intricate complexities of this cellular and developmental nexus (1, 34).

When targeting two paralogues together, the observed phenotypes of these double Crispants grossly phenocopy the individual Crispant with the most severe phenotype. However, in some instances, double Crispants have additive effects, such as shortened overall length (*lats1/2*) and smaller eye size (*ctgfa/b*) highlighting a potential functional redundancy. Notable exceptions are *nf2a/b*, where phenotypes of the double mutant are reversed compared with the single-mutant phenotypes that are observed in eye size (Fig 2E and F), a phenomenon also observed in *lats1/2* Crispants (Fig 2E and F). This might imply that a modest elevation of YAP/TAZ activity by moderately reducing the upstream kinase cascade activity, such as targeting *NF2*, or *LATS* individually (Fig 1A) increases organ size during development, whereas without the compensatory component from the other functional paralogue, elevated activity levels are likely detrimental. Overall, our combined analysis highlights the importance of the Hippo pathway in development, while emphasising context-dependent significance of distinct Hippo pathway components in the vertebrate.

## The Crispant screen reveals distinct regenerative roles of Hippo pathway components

As the Hippo pathway is integral to regeneration and repair (1, 75, 79, 80), we next sought to combine the advantages of the zebrafish regenerative capacity (1) and our efficient Crispant pipeline. This combination allows regenerative assays to be conducted within the timeframe of Crispant generation.

We decided to analyse two regeneration paradigms: mild and severe tail fin injury (Figs 3A and S2A), assumed to rely on distinct reparative molecular players. Both assays include excising part of the caudal fin in anaesthetised 72 hpf embryos, but with different amounts of the tail fin and notochord removed (Fig 3A). Mild tail fin regeneration involves regrowth of the fin fold epithelium, primarily because of compensatory cell proliferation. In contrast, severe tail fin regeneration occurs through a complex tissue

response, including the formation of a blastema, which requires dedifferentiation and proliferation (1, 81). Regeneration was analysed by imaging the caudal fins 48 h after transection (Figs 3A and B, S2, and S3). To compare regenerative growth with baseline growth rates, we analysed uninjured fin growth in parallel (Figs 3C and S2B). We noticed decreased fin growth in uninjured fins in *nf2a/nf2b*, *lats1/lats2*, and *ctgfb* Crispants, and an increased growth in *nf2a* and *lats2* Crispants (Fig 3C) highlighting component-specific functional differences. In addition, targeting one but not both upstream Hippo pathway (*nf2* and *lats*) paralogues leads to increased growth (Fig 3C). These data suggest that a single Crispant in both upstream inhibitory Hippo pathway components may be sufficient to enhance growth-promoting Yap1/Taz activity, but that double Crispants may, because of the lack of compensation, have excessive Yap1/Taz activity that prevents this growth-enhancing effect. However, we cannot fully rule out that these differential phenotypes might be caused by other defects, including those observed in the developmental analyses.

The baseline growth data allowed us to normalise regeneration to normal growth (Fig 3C). After mild tail fin injury, we observed that none of the Crispants show increased regeneration (Figs 3D and S3). *lats2*, *ctgfb*, and *yap1* Crispants show reduced regeneration after mild tail fin injury, whereas *lats1*, *ctgfa*, and *wwtr1* Crispants do not (Figs 3D and S3). Of interest, *yap1/wwtr1* double Crispants display a decrease in regeneration to the same level as *yap1* Crispant alone, highlighting that *wwtr1* does not play a significant role in this process (Fig 3D). Notably, *lats1/2* double Crispants have dramatically reduced regeneration.

We next evaluated regeneration after severe tail fin injury (Figs 3A, B, and E and S3). Interestingly, these analyses highlight marked differences in *yap1* and *wwtr1* Crispant phenotypes between the two regenerative paradigms. Surprisingly, *yap1* Crispants appear not to have defects. Notably, although *wwtr1* Crispants show only minor development defects compared with *yap1* Crispants (Fig 2), *wwtr1* Crispants display defects in severe tail fin regeneration (Fig 3E). However, unlike for mild regeneration, *yap1/wwtr1* double Crispants have more severe defects than either *yap1* or *wwtr1* alone, suggesting that *yap1* plays a partially compensatory role in this regenerative paradigm. Similarly, despite having no phenotypes in uninjured or mild injury conditions, *nf2b* Crispants exhibit a regeneration defect, which was exacerbated when *nf2a* was also mutagenised in the double Crispant, indicating compensation (Fig 3E).

*ctgfb* Crispants have increased severe tail fin regeneration (Fig 3E), despite having reduced general growth (Fig 3C) and regeneration in the mild tail fin experiments (Fig 3D). This reveals an

# Time course of mild tail fin injury regeneration

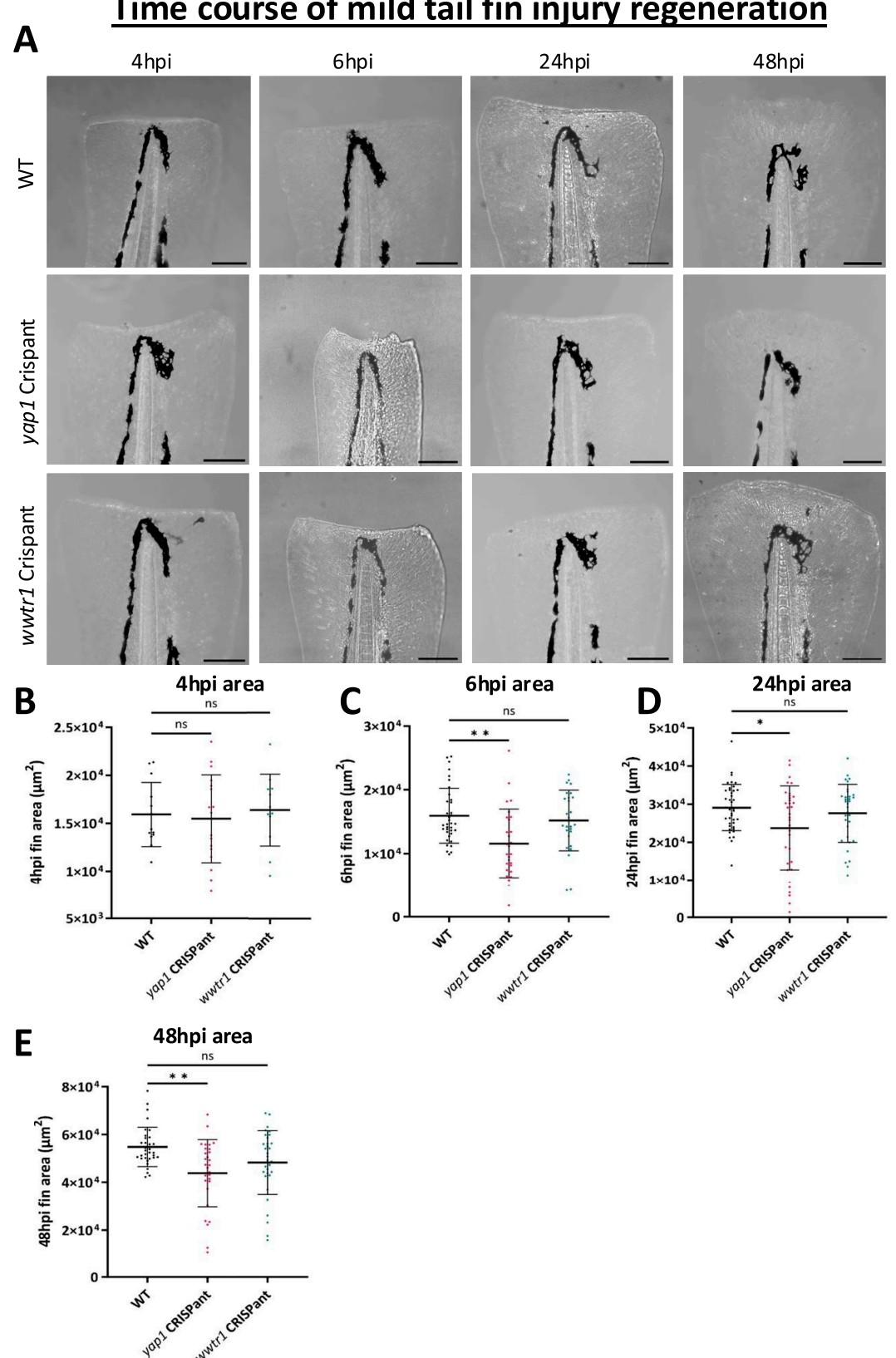

**Figure 4. Yap1 and Taz have distinct regenerative roles after mild tail fin injury.**
**(A)** Representative WT (top), *yap1* (middle), and *wwtr1* (lower) Crispant images of the regenerating tail fin after mild injury at 4, 6, 24, and 48 hpi. Scale bar represents 100 μm. **(B)** Quantification of fin area at 4 h post-mild tail fin injury. No change is seen in Crispants. **(C)** Quantification of fin area at 6 h post-mild tail fin injury. Fin area is significantly reduced in *yap1* Crispants only. **(D)** Quantification of fin area at 24 h post-mild tail fin injury. Fin area is reduced in *yap1* Crispants only. **(E)** Quantification of fin area at 48 h post-mild tail fin injury. Fin area is significantly reduced in *yap1* Crispants only. Mild tail fin injury regeneration is impaired in *yap1* but not *wwtr1* Crispants. Fin area is measured from notochord tip. Significance is analysed by one-way ANOVA (B, C, D) or Kruskal–Wallis test (E) with multiple comparisons. Each dot represents one fish. Results are from at least three independent repeats. Bars represent the mean ± SD. *P < 0.05; **P < 0.01; ns, not significant.

unknown (82, 83) inhibitory effect of Ctgfb in severe tail fin regeneration, contrary to the pro-regenerative role in mild tail fin regeneration (84) (Fig 3D). Notably, we do not observe a phenotype in the regenerating tail fin by solely targeting *ctgfa* (Fig 3D and E), but the combined loss of *ctgfa* and *ctgfb* reverses the phenotype observed in *ctgfb* loss. This suggests that altered Ctgfa activity might be partly responsible for the *ctgfb* Crispant phenotypes. Of interest, in zebrafish spinal cord regeneration Ctgfa plays an essential pro-regenerative role in the glial bridge (85, 86, 87). These complexities might underlie CTGF's opposing context-dependent roles observed in mammalian and clinical studies (82, 83, 84, 88), effects that are likely mediated through CTGF's prominent role in the extracellular matrix and cellular niche, where CTGF serves as a central multiparametric regulator of differentiation states, proliferation, angiogenesis, fibroblast activation, fibrosis, and cellular adhesion and migration (88, 89, 90, 91).

Crispant generation is highly efficient (Figs 1 and S1) but a relatively new methodology. We therefore sought to compare our Crispant results with stable mutants. A CRISPR-generated *yap1* stable mutant with a five-nucleotide deletion at 70 base pairs (bp) from the beginning of exon 1 and an 11-nucleotide deletion at 140 bp from the beginning of exon 2 was bred to homozygosity (Fig S4A and B). These mutations shift the reading frame and create a greatly disrupted amino acid sequence with premature stop codons (Fig S4C). Notably, the proportion of homozygous *yap1* stable mutants from heterozygous parents was lower than expected for a Mendelian ratio, whereas there was no change in heterozygote survival (Fig S4D). This indicates reduced homozygote survival, confirming previous reports (92). *yap1* stable mutants greatly phenocopy *yap1* Crispants in both development (Fig S4E–H) and regeneration (Fig S5A–D), with a general trend towards a slightly more severe phenotype (Figs S4E–H and S5A–D). For example, *yap1* stable mutants have a more stunted baseline tail fin growth (Fig S5B). This overall *yap1* Crispant and *yap1* stable mutant phenocopying emphasises the power of the Crispant approach. Notably, we observe defects in both mild and severe tail fin regeneration in *yap1* mutant larvae, which was not detected in the Crispant (Fig S5C and D). This initial apparent false-negative severe regenerative phenotype in the *yap1* Crispants highlights that caution is needed during experimental design and data interpretation when performing Crispant-based genetic screens; a feature that might be particularly relevant when severe developmental phenotypes are observed, as secondary phenotypes might be masked.

### Differential regenerative roles for the co-transcriptional mediators Yap1 and Taz

Next, we sought to further examine the phenotypes observed in our initial screen, focussing on the different tail fin injury regeneration phenotypes between *yap1* and *wwtr1* Crispants and resolving the difference between the *yap1* Crispant and *yap1* mutant in the severe injury model. In the screen, *yap1* Crispants show a reduction in regeneration after mild but not severe tail fin injury, whereas *wwtr1* Crispants have reduced regeneration after severe but not mild tail fin injury (Fig 3). Because we observed a defect in severe regeneration for *yap1* mutant larvae, we considered that *yap1* Crispants might also have reduced regeneration after severe tail fin injury, but that it was not detected in the screen because of potential sampling bias. It is plausible that the most severely affected developmentally disrupted *yap1* Crispants might die early and are therefore excluded from the initial analysis. We therefore designed a secondary Crispant phenotype analysis using both mild and severe tail fin injury model over multiple time points.

Here, we identified that during mild tail fin injury regeneration at 4 hpi, there is no change in fin area between WT and Crispants (Fig 4A and B). However, at 6 hpi *yap1* but not *wwtr1* Crispants have significantly reduced fin area compared with WT (Fig 4A and C) (67), suggesting the immediate epithelial response to fin removal is altered. *yap1* Crispants also have reduced regenerative outgrowth at 24 and 48 hpi (Fig 4A, D, and E), indicating a consistent defect in the whole regeneration process. *wwtr1* Crispants have no fin area change throughout the time course (Fig 4), recapitulating results obtained in the screen (Fig 3D). Therefore, Yap1 but not Taz (*wwtr1*) plays a critical role in resolving mild tail fin injury.

In the severe tail fin injury model, there is no change in the regenerated fin area between WT and both Crispants at 4 hpi (Fig 5A and B), whereas at 6 hpi, there is a reduction in the regenerative fin area in *yap1* but not *wwtr1* Crispants (Fig 5A and C). This difference is also observed at 24 hpi (Fig 5A and D), suggesting Yap1 is involved in the immediate tissue response to tail fin injury, recapitulating the observations from the mild tail fin injury model (Fig 4C–E). By 48 hpi, both *yap1* and *wwtr1* Crispants have reduced fin size (Fig 5A and E), recapitulating screen results (Fig 3E) indicating both paralogues are involved in tail fin outgrowth during severe tail fin regeneration and may therefore play non-redundant roles.

This more detailed analysis confirms the defect observed for *wwtr1* in the Crispant screen and indicates that *yap1* Crispants do have a severe tail fin regeneration defect. The findings therefore also emphasise the general importance of careful considerations in the experimental design, including secondary validations, when using Crispant screens as a discovery tool. Importantly, we also reveal that *wwtr1* Crispants have a specific regenerative defect after severe tail fin injury only, whereas *yap1* Crispants have a regeneration defect in both mild (Fig 4) and severe (Fig 5) injuries. The distorted developmental *yap1* defects add complexities to the interpretation of the regenerative *yap1* phenotypes, whereas the

# Time course of severe tail fin injury regeneration

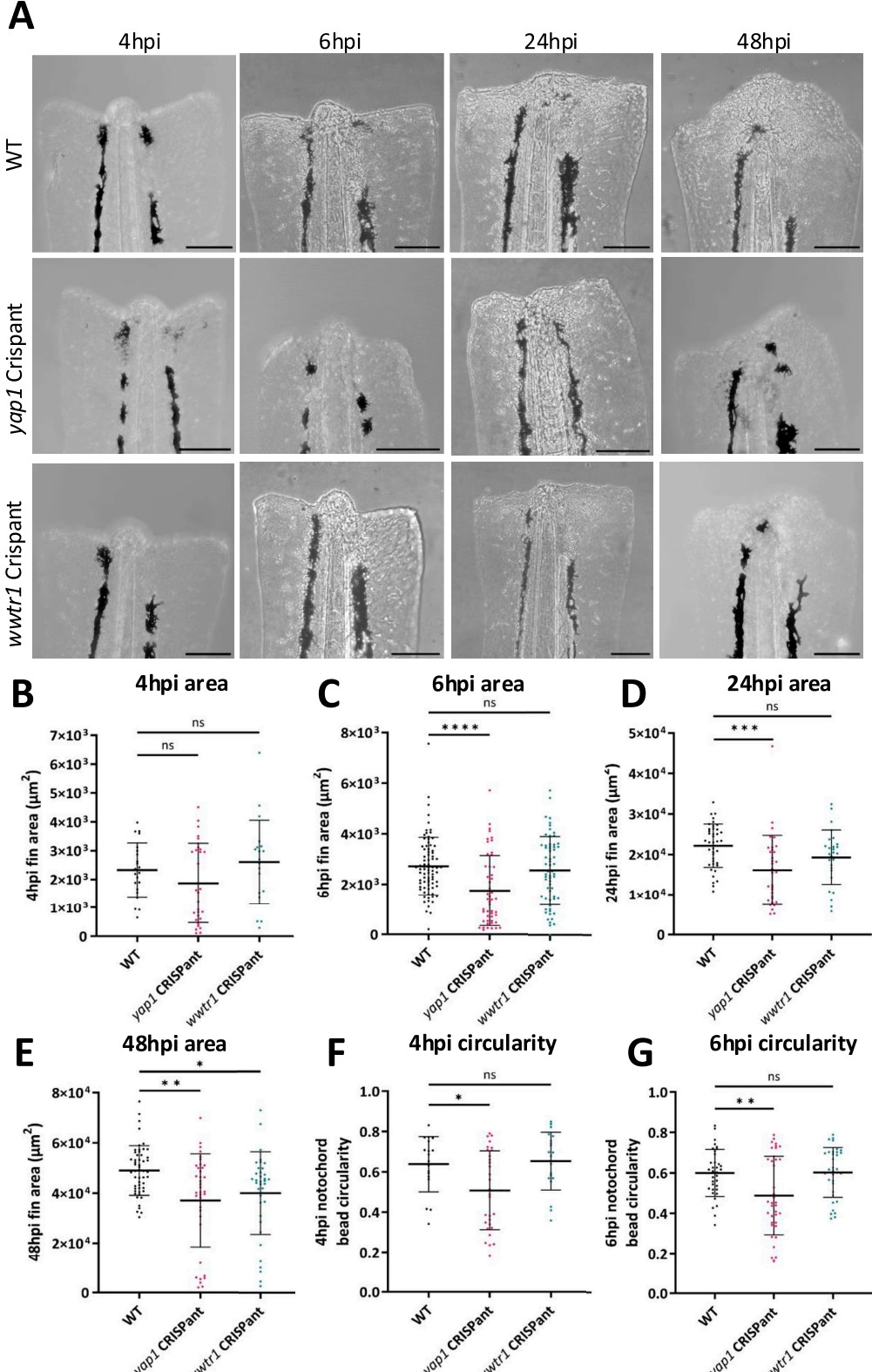

Figure 5. Yap1 and Taz have temporal distinct regenerative roles in regeneration after severe tail fin injury.

(A) Representative WT (top), *yap1* (middle), or *wwtr1* (lower) Crispant images of the regenerating tail fin after severe injury at 4, 6, 24, and 48 hpi. Scale bar represents 100 µm.
(B) Quantification of fin area at 4 h post-severe tail fin injury. No change is seen in Crispants. (C) Quantification of fin area at 6 h post-severe tail fin injury. Fin area is reduced in *yap1* Crispants only. (D) Quantification of fin area at 24 h post-severe tail fin injury. Fin area is reduced in *yap1* Crispants only.
(E) Quantification of fin area at 48 h post-severe tail fin injury. Fin area is significantly reduced in both *yap1* and *wwtr1* Crispants. (F) Quantification of notochord bead circularity at 4 h post-severe tail fin injury. Notochord bead circularity is significantly reduced in *yap1* Crispants only.
(G) Quantification of notochord bead circularity at 6 h post-severe tail fin injury. Notochord bead circularity is significantly reduced in *yap1* Crispants only. Fin area is measured in the notochord bead at 4 and 6 hpi, and from an approximate cut plane at 24 and 48 hpi. Significance is measured by the Kruskal–Wallis test (D, E) or one-way ANOVA (remainder) with multiple comparisons and significance calculated relative to WT only. Each dot represents one fish. Results are from at least three independent experimental repeats. Bars represent the mean ± SD. *P < 0.05; **P < 0.01; ***P < 0.001; ****P < 0.0001; ns, not significant.

# Temporal Yap/Taz activity during fin regeneration

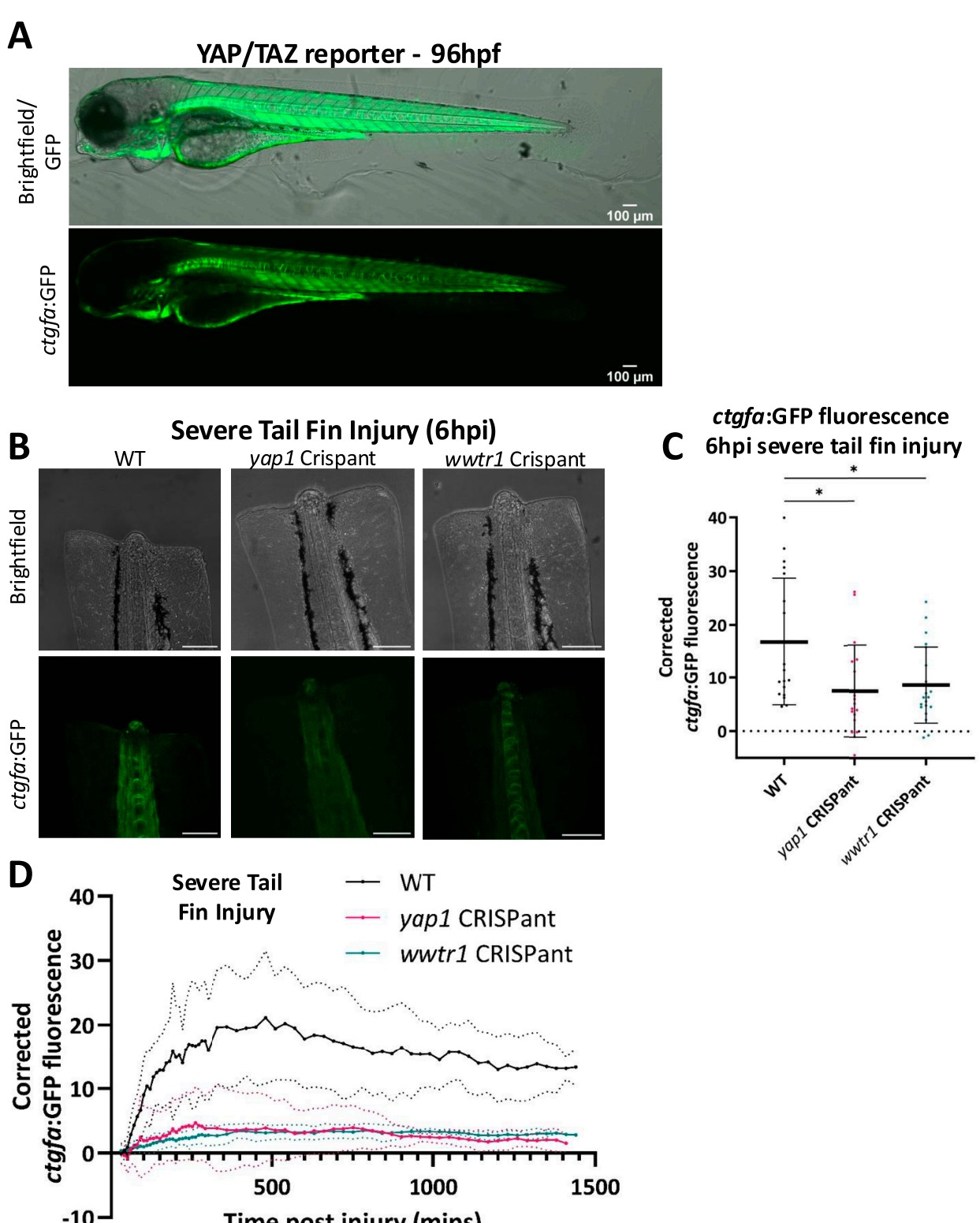

**Figure 6. Temporal Yap1/Taz activity during fin regeneration.**
**(A)** Sample confocal images of 96 hpf WT *ctgfa*:GFP larvae. **(B)** Representative images of *ctgfa*:GFP embryo tail fins at 6 h post-severe tail fin injury. **(C)** Quantification of mean corrected *ctgfa*:GFP fluorescence in the notochord bead at 6 h post-severe injury. GFP fluorescence is reduced in *yap1* and *wwtr1* Crispants. Each dot represents one fish. Bars represent the mean ± SD. Significance is calculated by one-way ANOVA with multiple comparisons. **(D)** Quantification of mean corrected *ctgfa*:GFP fluorescence in the notochord bead over 24 h post-severe injury. Fluorescence was reduced in Crispants relative to WT. A solid line represents the mean fluorescence for WT (black), *yap1* (red), and *wwtr1* (green) Crispants over time. A dotted line represents the SD for each genotype. Fluorescence is corrected to remove background and autofluorescence. *P < 0.05. Scale bars represent 100 μm.

regenerative defects in the *wwtr1* Crispants are specific (Fig 2). Notably, *wwtr1* Crispant severe tail fin injury defect is restricted to the later regenerative phase, whereas *yap1* Crispant defects are prominent from the early phase of regeneration.

A fundamental difference between mild injury, where only fin fold epithelium and mesenchymal tissue are removed, and severe injury is that severe injury involves removal of more complex tissues, including the notochord. The notochord is a hydrostatic organ, made up of a core of vacuolated cells, which are surrounded by sheath cells and function during development as the primary axial skeleton (56, 93, 94, 95, 96). After amputation, notochord cells immediately migrate to the injury plane and form a small "bead" that makes up the most distal part of the notochord. The notochord bead is an integral mechanotransductive signalling nexus, which is particularly prominent morphologically at the early stages of the injury (56, 93, 94, 95, 96). Given the role of Yap1 in tissue mechanics, we hypothesised that Yap1 and Taz might be mediating tail fin regeneration through regulating the notochord bead formation. We found that the circularity of the notochord bead deviates from WT in *yap1*, but not *wwtr1*, Crispants at 4 and 6 hpi (Fig 5F and G), suggesting that loss of Yap1 leads to aberrant notochord bead formation, possibly because of altered tissue mechanics (4, 31, 33, 97). This may contribute to the early onset of regeneration defects observed in the *yap1* Crispant. To delineate further mechanistic aspects, we sought to determine the patterning of the blastema using multiplexed HCR RNA-FISH (98). This allows us to visualise *col9a2*, *inhbaa*, and *aldh1a2*, each of which prominently localises to distinct regions of the blastema (93) (Fig S6A). Our analysis highlights distinct localisation of these blastema markers revealing no obvious localisation or structural changes in Crispants (Fig S6A), ruling out severe patterning defects.

## A Yap1/Taz reporter: *ctgfa*:dGFP reveals spatio-temporal Yap1/Taz activity during regeneration

The optical transparency of the embryonic zebrafish lends itself well to the use of transgenic fluorescent reporter lines and imaging (99). To gain further spatio-temporal insights into the role of the Hippo pathway, we replicated a *ctgfa* reporter fish. This approach is established on a previously characterised approach developed by the Link laboratory based on a published −1.0 kb ctgfa:d2GFP construct that allowed us to generate a *ctgfa:d2GFP* transgenic line (Fig 6A) (5, 6). Taking advantage of *CTGF*, a well-established YAP/TAZ-TEAD target gene (5, 6, 17, 70, 100), the destabilised GFP enables a real-time readout of Yap1/Taz activity with precise temporal resolution (5, 6). In the steady state, the reporter shows a strong signal in the heart, muscles, and the notochord, consistent with prior reports (5, 6) (Figs 6A and S7 and Video 1, Video 2, Video 3, and Video 4).

To visualise Yap1/Taz activity during regeneration within the same time window as analysed (Fig 5), we generated and compared individual *yap1* and *wwtr1* Crispants in the *ctgfa:d2GFP* background. As both *yap1* and *wwtr1* Crispants show a strong defect in the severe injury paradigm (Fig 3E), we measured GFP fluorescence during severe injury regeneration, initially focussing on 6 hpi. Given the notochord bead's function as a central signalling centre (56, 95, 96) and our observed altered morphology of notochord beads in *yap1* Crispants during severe tail fin injury regeneration (Fig 5F and G) (93, 94), we focussed on measuring GFP fluorescence changes within the notochord and notochord bead (Figs 6B and C and S7 and Video 1, Video 2, Video 3, and Video 4) (101, 102). At 6 hpi, GFP fluorescence is increased compared with uninjured tails in WT larvae, with high expression in the notochord and the notochord bead (Fig 6A and C). Fluorescence is markedly lower in both *yap1* and *wwtr1* Crispants (Fig 6B and C), suggesting both proteins are actively engaged in transcription activation of downstream target genes after injury. To visualise the temporal dynamics of Yap1/Taz activity, we performed confocal time-lapse imaging of the *ctgfa* reporter for 24 h post-severe tail fin injury in both WT and Crispants (Fig 6D). In WT larvae, GFP signal increases until ~8 hpi (480 mins), where it reaches the highest level and then slowly decreases and plateaus towards the end of the imaging period. This suggests that a rapid tissue response to wound damage involves immediate Yap1/Taz activation, followed by an ongoing moderate sustained Yap1/Taz activation during the regeneration process, as highlighted by the elevated *ctgfa* expression throughout regeneration. The *ctgfa* expression (Fig 6D) implies that Yap1 and Taz have dynamic transcriptional roles in the regenerative process.

We therefore next focussed our attention on the transcriptional changes happening during regeneration. We chose 24 hpi as our time point, as it is at the beginning of the regenerative outgrowth time window of severe tail fin regeneration (103). We therefore anticipate this time point captures key transcriptomic changes supporting regeneration. We compared the effect of mutagenising the co-transcriptional regulators *yap1* and *wwtr1* (17, 104) on the transcriptome at baseline and after severe tail fin injury. This allows us to review potential mechanisms mediating the defects that were observed in *yap1* and *wwtr1* Crispants. The principal component analysis (PCA) (Fig 7A) shows that samples generally cluster within their genotype and injury status. There is a large shift upon injury, highlighting, as expected, a dramatic effect of this injury paradigm (Figs 7A and B and S2A). At baseline, the *yap1* Crispant samples are more transcriptionally distinct from WT than *wwtr1* Crispants (Fig 7A), reflecting severe development defects observed in *yap1* but not *wwtr1* Crispants. This effect is lessened upon injury, suggesting that upon severe injury there is a large transcriptomic change in the embryo tail that is not solely driven by the transcriptional regulators Yap1 and Taz

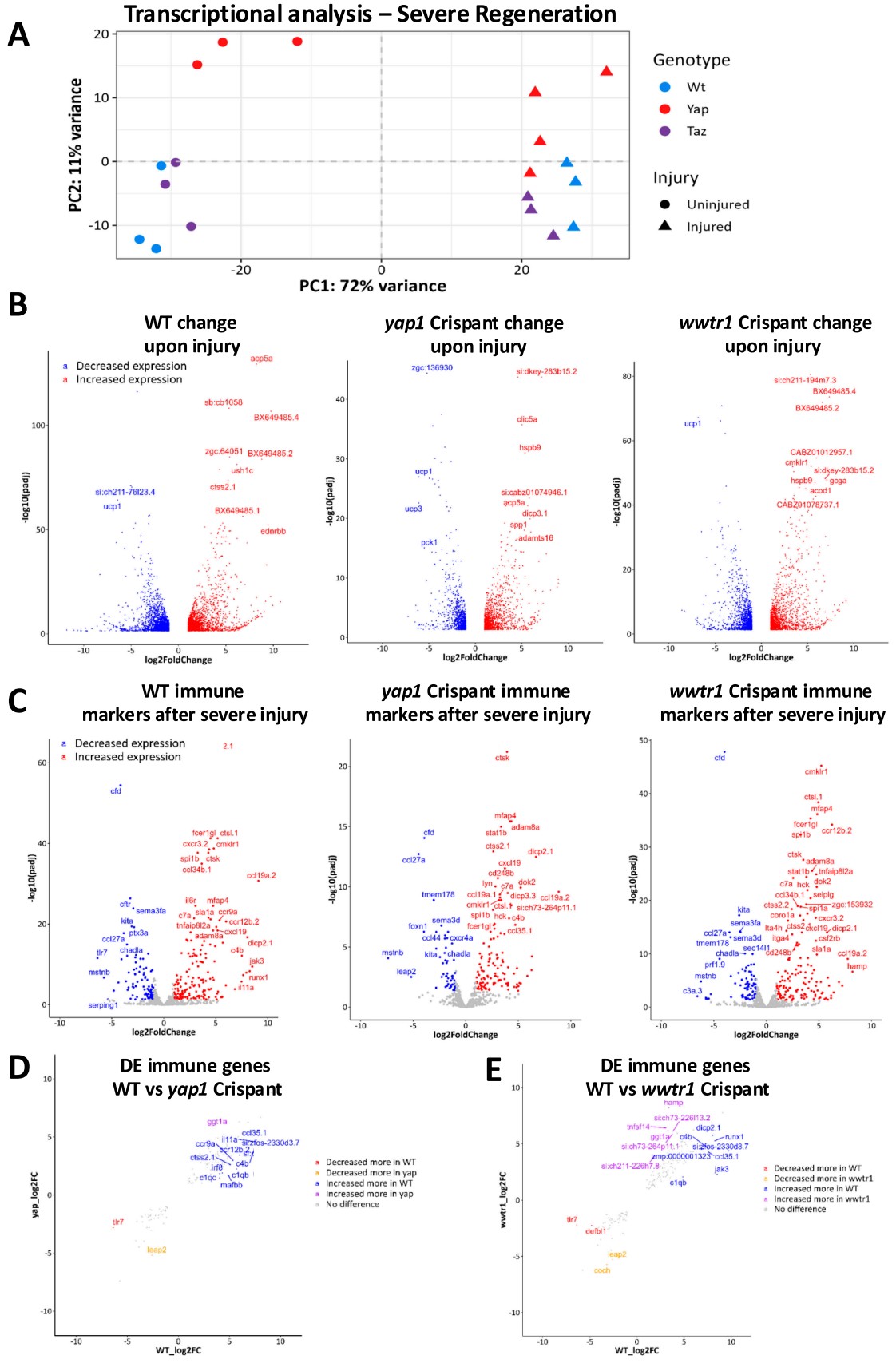

**Figure 7. Transcriptional changes during fin regeneration.**
**(A)** PCA plot of bulk RNAseq of fins from WT (blue), *yap1* (red) Crispant, and *wwtr1* (purple) Crispant at both baseline (circle) and 24 h post-severe tail fin injury (triangle). A large shift was seen upon injury. Note *yap1* Crispant samples varied more in uninjured samples than after injury. **(B)** Volcano plot depicting differentially expressed genes (DEGs) (blue down, red up) in 24 h post-severe tail fin injury WT (left) fish, *yap1* (middle) or *wwtr1* (right) Crispants. **(C)** Immune marker DEGs post-severe tail fin injury in WT (left) fish, *yap1* (middle), or *wwtr1* (right) Crispants. **(D)** Comparison between WT and *yap1* Crispants of immune DEGs after severe injury. **(E)** Comparison between WT and *wwtr1* Crispants of immune DEGs after severe injury. Cut-off for DEGs: $P$(adj) < 0.05 and log$_2$FoldChange > 1 or < –1. Cut-off for change between genotypes in (D, E) represents > 2 log$_2$FoldChange.

(Fig 7A). Notably, we observed that a range of the top most altered genes include chemokines and cytokines such as *cxcl18b* and *ccl19a.2*, suggesting that altered immune response might be contributing to the tail fin regenerative defects (86, 105, 106). We therefore analysed the differentially expressed gene (DEG) sets according to GO annotation "Immune-response" in more detail across the genotypes (Fig 7B and C). This highlighted robust gene expression changes upon reparative regeneration, and most were notably up-regulated in *wwtr1* and *yap1* Crispants (Fig 7C). To better understand genotype-specific differences, DEGs after injury in Crispants compared with WT were analysed within immune-response genes (Fig 7D and E). When comparing WT and *yap1* Crispants, there was a bias towards a greater number of genes being increased more in WT fish (Fig 7D). This suggests that *yap1* Crispants might have a more blunted response to injury than WT fish. This was not seen when WT fish were compared with *wwtr1* Crispants (Fig 7E).

These analyses emphasise changes in specific gene sets, including strong modulators of macrophage responses, a cell type critical for regenerative processes, highlighting likely immune cell–mediated regenerative phenotypes. Both macrophages and neutrophils respond to tail fin injury by migrating to the transection plane via a chemoattractant gradient (107). The short-lived neutrophils are the first responders, and neutrophils are the most prominent myeloid-derived cell type 6 h after injury, where neutrophils function as initial scavengers, where they are later succeeded by macrophages (108, 109, 110). We took advantage of fluorescent transgene fish lines expressing markers for macrophages (*mpeg*:GFP) and neutrophils (*lysC*-NLS:mScarlet) and analysed the abundance of macrophages and neutrophils in Crispants. To reveal the role of immune cells during regeneration, the macrophage and neutrophil numbers at steady state were examined in both the whole body and the caudal hematopoietic tissue (CHT). These data show that baseline macrophage and neutrophil numbers are lower in *yap1* Crispants, but not in *wwtr1* Crispants (Fig S8), highlighting a potential defect in a common myeloid progenitor upon *yap1* loss.

Macrophages but not neutrophils have well established pro-regenerative roles in resolving tail fin injury (105, 106, 107, 111, 112, 113). We therefore focussed on macrophages within the regenerative process in the severe tail fin injury paradigm, where loss of either Yap1 or Taz has functional consequences (Fig 5). Mammalian macrophages have historically been classified into two subtypes: the pro-inflammatory M1 macrophages and the anti-inflammatory M2 macrophages (114). Similar subtypes have been observed in the larval zebrafish, which have different roles during mild tail fin resolution (108, 113, 115). These dynamic macrophage

subpopulations are interconvertible and can be distinguished as M1-like macrophages that express *tnfα*, whereas M2-like macrophages do not (108, 116, 117). Macrophages appear to be recruited in a single wave to the wound, where they convert to M1-like macrophages, and then later switch both their shape and behaviour to M2-like macrophages (115). M1-like macrophages peak at 6 hpi, and then diminish during mild injury regeneration, whereas the number of M2-like macrophages peaks at 6 hpi and then plateaus (113). In addition, Tnfα and its receptor Tnfr1 are both necessary for fin regeneration via activation of blastema cells (113).

We investigated macrophage subtypes during severe injury regeneration using a *mpeg*:mCherry; *tnfα*:GFP transgenic line, where M1-like macrophages are co-labelled with both *mpeg* and *tnfα*, and M2-like macrophages are labelled with *mpeg* only. Representative images for severe tail fin regeneration are shown for 6 hpi (Fig 8A), 24 hpi (Fig 8B), and 48 hpi (Fig 8C). M1-like macrophage number increases during the initial 24 hpi, then plateaus, with Crispants having no significant deviation from WT (Fig 8D). In WT fish, M2-like macrophages steadily increase throughout the imaging period (Fig 8E). In *yap1* Crispants, M2-like macrophages are significantly reduced from 24 hpi, and in *wwtr1* Crispants, M2-like macrophages are decreased at 48 hpi. Notably, this temporal difference between *yap1* and *wwtr1* Crispants correlates with the regenerative phenotype observed at an earlier time point for *yap1* than *wwtr1* Crispants (Figs 5 and 8). This indicates that these Crispants may have a defect in the switch from M1-like to M2-like macrophages, suggesting that the defect in regeneration might be a result of an extended pro-inflammatory environment in the tail fin, including a lack of the pro-regenerative M2-like macrophages (105, 106, 107, 111, 112, 113, 118).

## Discussion

Our analysis highlights the substantial differences and molecular requirements needed to resolve the severe and mild tail fin injury paradigms. Our Hippo pathway-centric in vivo screen is the first of its kind, enabling us to reveal specific roles for individual Hippo pathway components in both regenerative processes and development. Our focussed pipeline allowed us to generate loss-of-function mutations in core Hippo pathway genes in zebrafish larvae, while analysing both developmental (Fig 2) and regenerative (Figs 3, 4, 5, 6, 7, and 8) phenotypes. These parallel phenotypic assessment studies allow, for the first time, an insight into the comparative importance of individual core Hippo pathway genes within the vertebrate lineage and highlight the functional diversity and complexities within the Hippo pathway.

# Macrophage recruitment severe fin regeneration

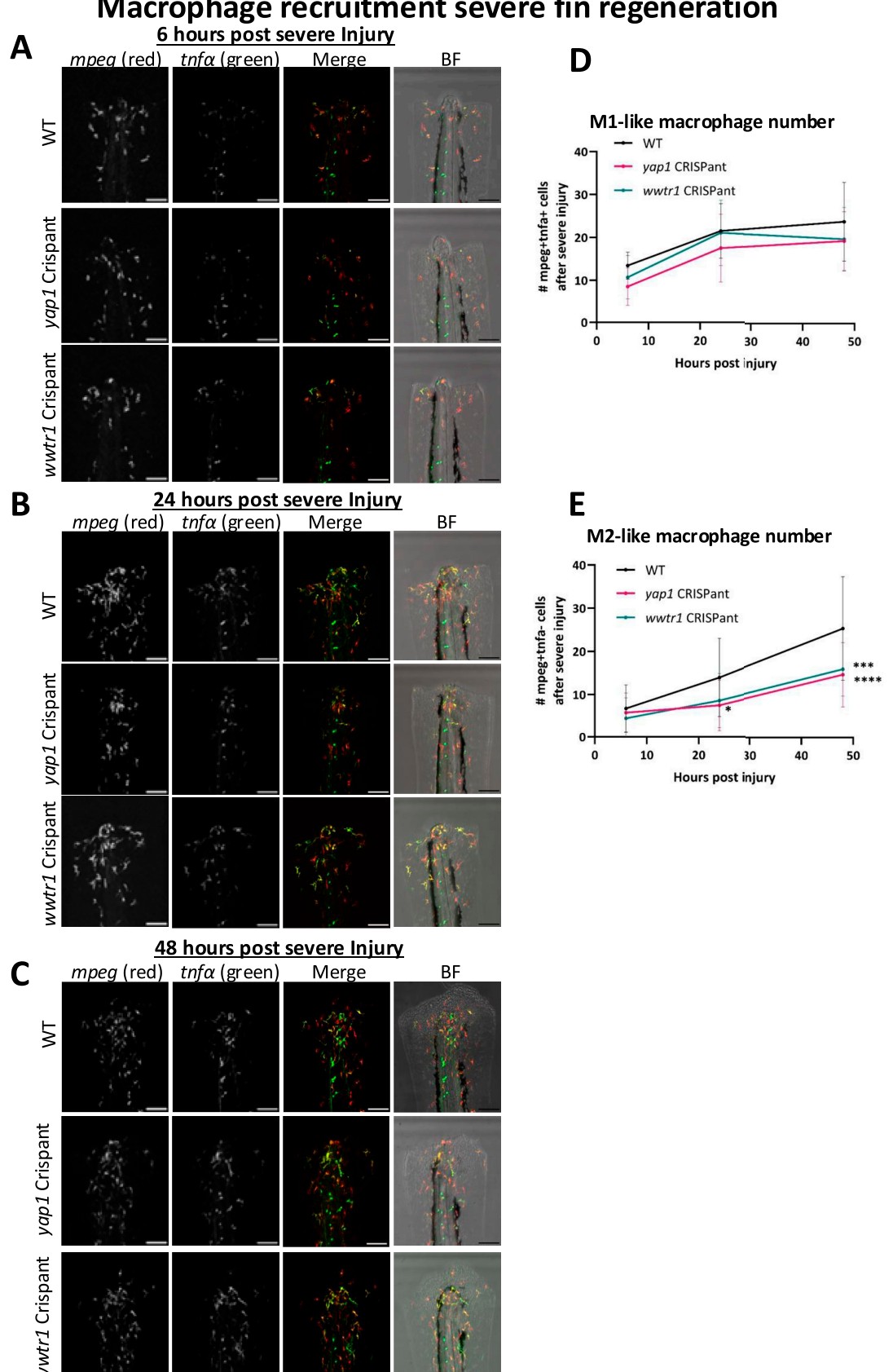

**A** 6 hours post severe Injury

**B** 24 hours post severe Injury

**C** 48 hours post severe Injury

**D** M1-like macrophage number

**E** M2-like macrophage number

**Figure 8. Macrophage subtype alterations during severe tail fin injury regeneration are differentially defective upon Yap1 or Taz loss.**
**(A)** Representative images of *mpeg* (*mpeg*:mCherry, red) and *tnfα* (*tnfα*:GFP, green)-positive macrophage infiltration in WT, *yap1* Crispant, and w*wtr1* Crispant fins at 6 hours post-injury (hpi) during severe tail fin regeneration. **(B)** Representative images of *mpeg*:mCherry and *tnfα*:GFP-positive macrophage infiltration in WT, *yap1* Crispant, and *wwtr1* Crispant fins 24 hpi. **(C)** Representative images of mpeg:mCherry and *tnfα*:GFP-positive macrophage infiltration in WT, *yap1* Crispant, and *wwtr1* Crispant fins 48 hpi. **(D)** Time course of M1-like (*mpeg+tnfα+*) macrophages during severe injury. Crispants do not significantly differ from WT at any time point. **(E)** Time course of M2-like (*mpeg+tnfα-*) macrophages during severe injury. *yap1* Crispants have reduced *mpeg+tnfα-* from 24 to 48 hpi, whereas *wwtr1* Crispants have reduced *mpeg+tnfα-* at 48 hpi only. Numbers are quantified from the start of the pigment gap to the end of the regenerate. Scale bar represents 100 $\mu$m. Bars represent the mean ± SD. Data are calculated from three independent repeats. Significance is calculated by two-way ANOVA, with multiple comparisons. *$P < 0.05$; ***$P < 0.001$; ****$P < 0.0001$.

Notably, Yap1 but not Taz loss causes a defect in regeneration after mild tail fin injury (Fig 4). Additional differential Yap1 and Taz regenerative roles are further highlighted by the earlier onset of regenerative defects in the severe tail fin paradigm upon *yap1* loss compared with *wwtr1* loss (Figs 5 and 8). Combined, our data highlight that *yap1* or *wwtr1* loss causes an overall reduction in the regenerative ability after severe tail fin amputation (Fig 3E), which might be partly caused by defective macrophage functionality (Figs 7 and 8) and altered transcriptional response (Figs 6 and 7), highlighted by the blunted *ctgfa*-GFP induction upon injury. This highlights that both Yap1 and Taz, in a functionally non-redundant manner, are required to maintain enhanced *ctgfa* expression throughout the regeneration process in resolution of severe tail fin injury (Fig 6). Notably, the onset of this regenerative defect occurs earlier in *yap1* Crispants than in *wwtr1* Crispants; these differential temporal defects are present as a defective notochord bead (4–6 hpi) (Fig 5F and G), followed by diminished M2-like macrophages (24 hpi) in the *yap1* Crispant (Fig 8E).

Challenges emerging during our studies included the complexities arising from studying a regenerative process in developing Crispants and the general lack of cell type–specific KO approaches in the zebrafish. Combined, these challenges limit the detailed molecular conclusions that can be drawn. Further multiparametric analysis, including single-cell approaches, examining regenerative paradigms in adult fish, and using further fluorescent reporter lines, will likely be productive to obtain a more detailed molecular understanding. Notably, most antibodies do not readily work for fish proteins, as antibodies overwhelmingly are developed with a focus on humans, and therefore raised against epitopes from human proteins. We were therefore limited in the ability to fully establish the loss of protein in all Crispants across all the genes targeted. However, as we validated the approach for Yap1 and Taz, where antibodies recognising fish Yap1 and Taz are available (Fig 1), combined with the conservation of our general pipeline, we are confident that our approach in general causes loss of targeted protein. We used the same efficient (two-guide) Crispant approach in targeting genes in our screen (Figs 1 and S1), but as we did not directly test the protein levels, we cannot fully rule out the expression of some remaining proteins in these, potentially masking phenotypes. Importantly, our more detailed functional focus centred on the evaluation of Yap1 versus Taz loss.

The recent development of compounds that activate YAP (and potentially TAZ) and that are found to drive regeneration across multiple tissues (65, 119, 120) put our findings into further context. Importantly, these compounds target various components and nodes of the Hippo pathway. It is therefore critical to identify the consequences of targeting different individual Hippo pathway components in vivo. These detailed insights are needed to fully understand the therapeutic potential and possible detrimental effects on homeostasis of the individual signalling nodes within the Hippo pathway. The importance of these analyses is further clarified by the widespread realisation that YAP and TAZ hyperactivation plays key roles in multiple cancer types (45, 70, 77, 79, 121, 122), including in the frequent *NF2* loss-of-function mutations in both pleural mesothelioma, a deadly cancer originating in the mesothelial lining of the lung (70, 123, 124, 125), and schwannomas, tumours of the nervous system (126, 127, 128, 129). Gaining precise mechanistic insights will enable selective activation of regenerative programmes without triggering tumorigenic outcomes. This knowledge can guide the development of safe, targeted interventions that enhance tissue repair, modulate cell fate, and restore function after injury, while maintaining the essential growth-suppressive safeguards that protect against malignancy. Combined, this highlights the need for a cautious well-informed approach to fulfil and facilitate this substantial translational regenerative potential (65, 119, 120).

Our initial characterisation, cataloguing, and discoveries provide a key milestone in the understanding of individual Hippo pathway components in the vertebrate lineage.

# Materials and Methods

A table with the main materials used in this study is included at the end of this section.

### Crispant generation

CRISPR/Cas9 guide RNAs were designed using gene sequences imported from Ensembl into Benchling software. Benchling was used to identify predicted efficient CRISPR/Cas9 guide RNAs, which were then validated and ordered via IDT (custom or predesigned Alt-R CRISPR-Cas9 gRNA, cRNA 10 nmol) (for sequences, see Table 1). Primers used to determine guide RNA cleavage were designed using NCBI Primer-Blast (130) with sequences imported from Benchling. Primers were chosen with a Tm of 52–58°C, between 150- and 250-base pair (bp) amplicon length, the putative guide RNA cleavage site being as close to the centre of the amplicon as possible, and without including the cleavage site of any co-injected guide RNAs (primer sequences shown in Table 2). The latter parameter was not applied when guide RNAs targeted different restriction enzyme (RE) digest sites (excluding T7 endonuclease I (T7E1) digest).

A microinjection mix was generated with all reagents diluted in 1:1 dH$_2$O:phenol red to aid visualisation of injection. Components were mixed and incubated for 5 min at RT. Three microinjection groups were analysed: Cas9-only–injected fish (Cas9 and tracrRNA), gRNA-only–injected fish (gRNAs and tracrRNA), and embryos injected with the full Cas9-gRNA complex (Cas9, gRNAs, and tracrRNA). Mixes were injected at 2 nl into one-cell-stage embryos (1,000 pg Cas9, 50 pg of each guide RNA, and 200 pg tracrRNA). Needles were pulled from a glass capillary using a Flaming/Brown Micropipette puller P-97 (Sutter Instruments) using the following settings: heat = 580; pull = 250; vel = 50; time = 125.

Microinjection mix was loaded into the needle using Eppendorf microloader tips and mounted onto a PV820 PicoPump micro-injector rig (WPI). Injection parameters were optimised to give a bolus of the desired size (using $V = \frac{4}{3}\pi r^3$, where V = volume and r = radius) on a 1-mm graticule with a droplet of mineral oil, viewed through a light microscope. Embryos were lined along the edge of a glass microscope slide in a petri dish lid. 2 nl of the CRISPR mix was injected into the yolk of the single zygote cell, as close to the cell as possible without damaging it. Resulting embryos, as well as stable mutants, were used for further analyses.

## Stable mutant generation

To generate stable mutants, Crispants (F0) were grown to adulthood. Once breeding age was reached, F0 fish were crossed with WT fish and half of the offspring were genotyped in bulk where possible to confirm that the F0 founder contained a mutation. The remaining offspring (F1) were grown to adulthood, then genotyped individually using PCR and Sanger sequencing. Heterozygotes of interest were then outcrossed (bred with a different strain) to generate F2 offspring, which were genotyped using DNA obtained from fin clips at adulthood to identify F2 heterozygotes. These F2 heterozygotes were incrossed (bred with siblings), and the resulting F3 offspring were grown to adulthood. DNA was extracted from fin clips, and homozygous F3 fish were identified by PCR (see below).

## gDNA sequencing of F2 fish

Fin clipping of adult fish was performed in accordance with Home Office Regulations. gDNA was extracted from fin clips, and Taq PCR (PCR settings: 95°C for 30 s, 30 cycles at 95°C for 15 s, 50°C for 15 s, 68°C for 30 s, and final extension at 68°C for 5 min) was performed using primers shown in Table 3, designed to amplify an ~150-bp-long region around the CRISPR target site. These sequences were then inserted into a pSC vector, transformed into bacteria using SOC medium, and amplified according to StrataClone PCR Cloning Kit instructions. Positive colonies were selected for overnight liquid culture, from which DNA was extracted using an E.Z.N.A. Plasmid DNA Mini Kit I, diluted to 100 ng/µl, and sent to Source BioScience for Sanger sequencing with an M13R primer. Sequences were cleaned in the FinchTV app and aligned to published sequences from Ensembl using Clustal Omega (ebi.ac.uk/Tools/msa/clustalo).

## cDNA sequencing

Sequencing of cDNA of the *yap1* gene was performed on F1 heterozygotes to identify the full mutant Yap1 sequence. Fin clipping of adult fish was performed in accordance with Home Office Regulations. Two primer sets were used, one to amplify the first half of the *yap1* gene (A), and another to amplify the second (B) (Table 4). Sequences were then inserted into a pSC vector, transformed, and amplified, and DNA was extracted and diluted as described above. Samples were sent to Source BioScience for Sanger sequencing with M13R and T7F primers. Sequences were cleaned in the FinchTV app and aligned to published sequences as above.

## Homozygous fish screening

F3 offspring (mix of WT, heterozygotes, and homozygotes) were grown to adulthood. Fin clipping of adult fish was performed in accordance with Home Office Regulations, and gDNA was extracted (see below). For *yap1* mutants, PCR and T7E1 digestion was performed using the *yap1* guide 1 primer described in Table 2. This identified heterozygotes. The remaining fish were outcrossed with WT fish, and the resulting embryos underwent gDNA extraction, PCR, and T7E1. T7E1-positive embryos were assumed to come from homozygous parents (as an outcross would give heterozygous offspring).

## *ctgfa*:dGFP transgenic reporter fish

A Tol2-*ctgf*(−1 kb)-d2GFP-pA (referred to throughout as *ctgfa*:GFP) reporter plasmid was kindly donated by Prof. B. Link (Medical College of Wisconsin) (5, 6). 25 ng/µl plasmid was co-injected into 1-cell stage embryos as a 1-nl bolus with 25 ng/µl Tol2 mRNA diluted in phenol red. F0 embryos were screened at 48 hpf, and embryos with GFP fluorescence in the heart and trunk were raised to adulthood. These F0 were then outcrossed with WT fish, and GFP-positive F1 embryos were again screened. Batches where 50% F1 embryos were GFP-positive were raised to adulthood before themselves being outcrossed with WT fish, screened for 50% heart and trunk fluorescence at 48 hpf (F2), and raised to adulthood. This approach was repeated to generate F3 adults that were assumed to contain a single stable GFP insertion. These F3 adults were then incrossed and the F4 GFP-positive offspring (heterozygotes and homozygotes) raised to adulthood. F4 homozygous adults were identified by outcrossing with WT fish whereafter the offspring was screened to identify the percentage of GFP-positive embryos, with homozygotes giving 100% GFP-positive F5 offspring, and heterozygotes giving 50% GFP-positive F5 offspring.

## gDNA extraction

Whole embryos at 24 hpf (for RE analysis) or fin clips from adult fish (for DNA sequencing) were incubated at 95°C for 20 min in 50 mM NaOH. 10% 1 M Tris was added before tubes were spun down at 14,000g for 10 min, and the supernatant was transferred to a fresh tube. Volumes added were dependent on the number of embryos from which DNA was extracted. For 1 embryo (analysis of guide cleavage efficiency), 50 µl NaOH/5 µl Tris was used, and for more embryos and fin clips, 100/10 µl was used.

**Table 1. CRISPR/Cas9 guide RNA sequences.**

| Gene | Ensembl ID | Guide | Sequence (5′–3′) |
|---|---|---|---|
| *ctgfa* | ENSDARG00000042934 | *ctgfa* guide 1 | GCUUACACCAGGUGAACACU |
| | | *ctgfa* guide 2 | CUGUGCAGACACCAAUACGA |
| *ctgfb* | ENSDARG00000104292 | *ctgfb* guide 1 | CAGGGAUCUGGACACGUCGT |
| | | *ctgfb* guide 2 | CCACGACGUGUCCAGAUCCC |
| *cyr61* | ENSDARG00000023062 | *cyr61* guide 1 | CGCUGCAGACCCACUCCUGG |
| | | *cyr61* guide 2 | CUGCGACCACAGCAAGGGCC |
| *lats1* | ENSDARG00000003751 | *lats1* guide 1 | CAACAGGAUAAACUUCCAGG |
| | | *lats1* guide 2 | GCCCACGACAGCUCCUCCAG |
| *lats2* | ENSDARG00000078864 | *lats2* guide 1 | UAGUUGUUAAACUUCUGGGG |
| | | *lats2* guide 2 | CUACAGGAUCUGGUCAAUGC |
| *nf2a* | ENSDARG00000020204 | *nf2a* guide 1 | AAAUGGGACUCUCUCUUCGG |
| | | *nf2a* guide 2 | AACACUAUAGGCUCCUCUUU |
| *nf2b* | ENSDARG00000025567 | *nf2b* guide 1 | GACCUGGUAUGCCGGACCAU |
| | | *nf2b* guide 2 | UAAAGUCAAAGUCAGCACUA |
| *wwtr1* | ENSDARG00000067719 | *wwtr1* guide 1 | UGGCGACAUGGAUCACCUGG |
| | | *wwtr1* guide 2 | CCGGCACCAGUCCUGCGAUG |
| *yap1* | ENSDARG00000068401 | *yap1* guide 1 | GUUAAAAAGAGCCUCCAGAU |
| | | *yap1* guide 2 | GAUGACAUGCCGCUGCCCCC |
| Synthetic crRNA sequence (IDT standard) | | | GUUUUAGAGCUAUGCUGUUUUG |

### Restriction enzyme analysis of CRISPR guide efficiency

gDNA was amplified using PrimeSTAR Max Premix (PCR settings: 30 cycles at 98°C for 10 s, 55°C for 5 s, 72°C for 10 s) with primers as shown in Table 2 (correct amplification was confirmed by running on a 1% agarose gel), then digested using various restriction enzymes. Different parameters and buffers were used depending on the restriction enzyme used for digestion (Table 5). Once digested, DNA was run and separated according to size on a 1.5–2% agarose gel containing gel red and visualised using a Gel Doc.

### RNA extraction

Uninjured and injured samples were collected from 50 embryos per sample. For injured samples, fish were injured at 3 dpf using the severe tail fin injury paradigm, then left to recover for 24 h in 0.3X Danieau's solution (diluted from 30X: 1740 mM NaCl, 21 mM KCl, 12 mM MgSO$_4$·7H$_2$O, 18 mM Ca(NO$_3$)$_2$, 150 mM Hepes). Both uninjured and injured tail fins were removed, caudal to the end of the dorsal vein (at the beginning of the pigment gap). Tissue was collected in chilled PBS and kept on ice for a maximum of 1 h before extraction. RNA extraction was performed using the QIAGEN RNeasy micro kit, and initial quality check and sample concentration were determined by NanoDrop.

### Western blotting

Sample preparations were as follows: Embryos pooled and used per sample were 24 hpf—25; 48 hpf—20; 72 hpf—10; 96 hpf—10; 120 hpf—20. Embryos were deyolked using a protocol adapted from reference 131. Embryos were dechorionated if needed, then mixed with 100 µl deyolking buffer (1/2 concentration of Ginzberg's fish Ringer's solution), and homogenised by vortexing. Samples were spun down, and the supernatant was discarded before being dissolved in 50 µl 2X SDS–PAGE loading buffer and incubated at 95°C for 5 min. Samples were spun down again, followed by transfer of the supernatant to a fresh tube. 10 µl samples were loaded into a SDS-containing 10% polyacrylamide gel, proteins separated by electrophoresis and transferred onto a nitrocellulose membrane. Membranes were incubated in Ponceau solution for 5 min, imaged, then destained in 5% milk/PBS at RT for a minimum of 30 min before incubation in primary antibody (1:1,000 in 5% BSA) overnight at 4°C. Membranes were then washed three times in TBST, probed with an HRP-conjugated secondary antibody (1:10,000 in 5% milk) at RT for 1.5 h, then washed again. Proteins were visualised using Immobilon reagent onto X-ray films in a dark room developer. Quantification of Western blots was carried out in ImageJ/Fiji (132). Loading levels were determined by analysing the peak height between ~55 and 100 kD (where possible) of a Ponceau blot. Values for bands of interest were then normalised to loading values, and these normalised values for the injected groups were then normalised to WT values and plotted for each independent replicate.

### 96 hpf imaging and quantification

96 hpf embryos were lightly anaesthetised using tricaine, mounted with a left (lateral) view in 2–3% methylcellulose, and imaged in

**Table 2. CRISPR/Cas9 guide RNA analysis primers.**

| Construct | FW/REV | Sequence (5'-3') | WT product length (bp) |
|---|---|---|---|
| ctgfa guide 1 | FW | TGGAGCTCTTTAGCAAATGT | 150 |
| | REV | CCCACAGGTGTCCAGAA | |
| ctgfa guide 2 | FW | AGAACGAGATGTTTGCGAC | 160 |
| | REV | AAGATTGTTTACAATTCTGACTCT | |
| ctgfb guide 1 and 2 | FW | GATCTGGGTGGTGTGATTTA | 220 |
| | REV | GACTGGAAGGTGTCTTCTTG | |
| cyr61 guide 1 | FW | TATCCTCCAGCTGCCC | 180 |
| | REV | CCCCATAGTTGCACTCCA | |
| cyr61 guide 2 | FW | GTGCTCCTGTGCTCC | 150 |
| | REV | GTGGCTGGCCCCATA | |
| lats1 guide 1 | FW | ATGTGATCAATGCAATGTGG | 160 |
| | REV | TTAATAATGCCTACCCGCTT | |
| lats1 guide 2 | FW | GATGAGTCCAGACAGCAA | 190 |
| | REV | CTGCATTACCTCATCAAAGC | |
| lats2 guide 1 | FW | GAGTCAGGCACTTCATGG | 180 |
| | REV | ACCTGAATCATTGGCATAGG | |
| lats2 guide 2 | FW | ATGCCAATGATTCAGGTCC | 195 |
| | REV | TCAAGTTACTGTTGACTACAATTC | |
| nf2a guide 1 | FW | TGTAGCTTTTAAGCAGAGAGG | 210 |
| | REV | GAACTCCAAATCAGCATCCA | |
| nf2a guide 2 | FW | AAGCAGAGTGCAACACTTAT | 190 |
| | REV | AAAATCATCCTGAGCCAGTG | |
| nf2b guide 1 | FW | GTGTCTGTCTGATTCTGTGT | 230 |
| | REV | TGGGAACAGATTTTAGGCTC | |
| nf2b guide 2 | FW | AATTATTTGTGTTTTCAGGTGT | 175 |
| | REV | GGACACAGAATCAGACAGAC | |
| wwtr1 guide 1 | FW | AGTTTGTTTTGACCATGAGC | 150 |
| | REV | GCATATCCTTGTTCCTCCAG | |
| wwtr1 guide 2 | FW | GTTCTCTCCCGCCGAG | 200 |
| | REV | GAGAAAGTACTTCTGGCCGT | |
| yap1 guide 1 | FW | GGTGTTTTTGGCAAGCAGCA | 400 |
| | REV | GGAGTGGGACCTTGGCTCTG | |
| yap1 guide 2 | FW | CAGATGCAGGTACTGCTGGTA | 440 |
| | REV | TGCCTTCATCATGTAAATGAACTGC | |

**Table 3. gDNA sequencing primers.**

| Construct | FW/REV | Sequence (5'-3') |
|---|---|---|
| yap1 guide 1 | FW | GAGGGCTGTGATTTTCAATG |
| | REV | GCAGGAAAAGTTTGTGGAAA |
| yap1 guide 2 | FW | CCTGTGGGGTATTTGCTTTA |
| | REV | TGCAGTTTACACAGAGTTGA |

**Table 4. cDNA sequencing primers.**

| Construct | FW/REV | Sequence (5'-3') |
|---|---|---|
| yap1 | FW A | ATGGATCCGAACCAGCA |
| | REV A | TTGTTTCCACTCATCACTCC |
| | FW B | GAATTTCCCAGAGTGCCC |
| | REV B | CTATAGCCAGGTTAGAAAGTT |

**Table 5. Restriction enzymes used and parameters for guide RNA efficiency analysis.**

| Restriction enzyme | Buffer | Guide cleavage site targeted | Parameters |
|---|---|---|---|
| BstXI | 3.1 | *ctgfb* guide 2 | Digestion at 37°C for 15 min |
| Bsu36I | CutSmart | *tp53* guide 1 | Digestion at 37°C for 15 min |
| EcoO109I | CutSmart | *cyr61* guide 2 | Digestion at 37°C for 15 min |
| Hpy99I | CutSmart | *ctgfb* guide 1 | Digestion at 37°C for 1 h |
| PfoI | Tango | *cyr61* guide 1 | Digestion at 37°C for 1 h |
| SacI | 1.1 | *tp53* guide 2 | Digestion at 37°C for 15 min |
| SexAI | rCutSmart | *wwtr1* guide 1 | Digestion at 37°C for 1 h |
| T7E1 | 2 | *ctgfa* guide 2 | DNA annealed at 95°C for 5 min, then reduced to 25°C over 20 min. Subsequent digestion at 37°C for 15 min |
| | | *lats1* guide 1 and 2 | As above |
| | | *lats2* guide 1 and 2 | As above |
| | | *wwtr1* guide 2 | As above |
| | | *yap1* guide 1 and 2 | As above |
| | | *nf2a* guide 1 and 2 | As above |
| | | *nf2b* guide 1 and 2 | As above |
| XcmI | 2.1 | *ctgfa* guide 1 | Digestion at 37°C for 1 h |

bright-field using a Leica M205 FCA microscope and LAS X software. Images were quantified in ImageJ by outlining areas of interest and calculating length/area as required.

Embryo length was quantified by measuring the length of a line manually drawn along the length of the embryo's body from the tip of the head, along the notochord, and ending at the tip of the body (excluding the tail fin). Standard length was not used because of the twisting of the body of some Crispants, which would have unintentionally skewed length measurements.

The swim bladder area was quantified using the circle tool in ImageJ (132) to outline the swim bladder, and the area of the circle was measured. Eye size was quantified using the circle tool in ImageJ to outline the embryo eye on the left (lateral) side, and the area of the circle was measured.

## Tail fin regeneration measurements in the CRISPR/Cas9 screen

72 hpf embryos were anaesthetised using tricaine, and a portion of the caudal tail was excised using a sharp scalpel. Embryos were imaged, then separated into individual wells of a 24-well plate, and left for 48 h to recover before imaging again. The same process was followed for analysis of baseline growth, except without caudal fin excision. The area of tail fin was measured, and fin growth was compared between individual fish (Fig S2A). If embryos died or were dying, the embryo was excluded from statistical analyses.

## Tail fin regeneration for in-depth tracking

72 hpf embryos were anaesthetised using tricaine, and a portion of the caudal tail was excised using a sharp scalpel. Embryos were left to recover in 0.3X Danieau's solution. Regenerated fin area was measured at 4, 6, 24, and 48 hpi after both mild and severe tail fin injury.

## Live imaging (single time points)

Embryos were anaesthetised using tricaine in 0.3X Danieau's solution and mounted in a glass-bottomed imaging chamber using heated 1% low melting point agarose. When set, agarose was covered with 0.5X strength tricaine in 0.3X Danieau's solution. Samples were imaged live using a Leica Sp5 confocal microscope (20X air objective) ensuring no pixel saturation.

## Live developmental imaging (*ctgfa*:GFP)

To image *ctgfa*:GFP fish, these fish were crossed, and their embryos were incubated in 0.003% PTU at 0 dpf to stop melanocyte development, and ease imaging, and 2, 3, 4 and 5 dpf, embryos were placed individually in a 96-well Grainer imaging plate in 0.3X Danieau's solution with 0.003% PTU and tricaine. Fish were aligned in agar moulds, which were embedded into a 96-well plate before imaging using a 3D-printed orientation tool (133). These images were acquired on the Opera Phenix Plus high-content imaging system through 300 $\mu$m of the embryo depth. Images were processed in ImageJ/Fiji (132) where brightness and contrasts were adjusted individually for each image.

## Live imaging (time lapse)

Embryos were anaesthetised using tricaine in 0.3X Danieau's solution and mounted in a glass-bottomed imaging chamber using heated 0.7% low melting point agarose. When set, agarose was removed from the tail of the embryo using a scalpel. Fish were then covered with 0.25X strength tricaine in 0.3X Danieau's solution.

Samples were imaged live at 28°C using a Leica Sp5 confocal microscope with a Leica HC Plan Apo 20x air objective (NA 0.70 Ph2) ensuring no pixel saturation. Images were initially acquired every 10 min from 30 min post-injury to ~5 h post-injury, when images were acquired every 30 min until 24 h post-injury. GFP fluorescence was quantified in the central blastema, then corrected for fluorescence in a non-regenerating area of the fin (dorsal to the caudal vein), and for autofluorescence in a non-GFP fish.

## Whole-body imaging for macrophage subtypes

Tg(*lysC*-NLS:mScarlet) (ZDB-TGCONSTRCT-211026-2), tgBAC(*tnfα*:GFP) (134), Tg(*mpeg*1.1:eGFP) (ZDB-FISH-150901-5341), and tg(*mpeg*1.1:mCherry) (135) fish were incrossed if needed. Resulting embryos were incubated in PTU from 0 dpf, with microinjection of *yap1* or *wwtr1* CRISPR as required. At 3 dpf, embryos were screened and placed individually in a specialised 96-well imaging plate (ZF plate, Funakoshi) in 0.3X Danieau's solution with 0.003% PTU and tricaine. Fish were aligned into imaging windows just before imaging, by spinning down at 100*g* for 1 min followed by, if needed, manual adjustments. Embryos were imaged using the Opera high-throughput imaging system, using automatic detection of the whole body. Images were taken through 300 *μ*m of the embryo depth (z-stack size = 1 *μ*m). Images were stitched and processed to quantify cell numbers in the whole fish. Quantification of cells in the CHT was performed manually in QuPath (136), with the CHT defined from the end of the yolk extension and reaching along the trunk of the embryo to the start of the pigment gap.

## HCR RNA-FISH

Embryos underwent severe tail fin injury at 3 dpf and fixed at either 24 or 48 hpi. Uninjured embryos were fixed at 96 or 120 hpf. Fixation was performed in PFA overnight, followed by dehydration, then permeabilisation in methanol for storage before rehydration in PBST, treatment with proteinase K (8 *μ*g/ml for 15 mins), and post-fixation with intermediate wash steps. Samples were then pre-hybridised in probe hybridisation buffer for 30 mins at 37°C before addition of the probe solution and incubation overnight at 37°C. Multiple washes in probe wash buffer and SSCT were then performed, followed by pre-amplification in amplification buffer for 30 mins at RT, whereas the HCR hairpins were heated to 95°C for 90 s and cooled to RT in the dark. Hairpin solution was then added to the samples and incubated overnight at RT in the dark. Hairpin solution was removed by washes in SSCT and mounted for imaging using an antifade mountant. Imaging was performed on a Leica Sp8 confocal microscope. Spectral unmixing using preset Alexa fluorophores was performed, where required, to separate the fluorophores. HCR RNA-FISH reagents were from Molecular Instruments. Targets and amplifiers were *aldh1a2* mRNA (B4-594), *col9a2* mRNA (B1-514) and *inhbaa* mRNA (B2-488).

## RNAseq analysis

Samples were submitted for RNA sequencing to the Genetics Core, Wellcome Trust Clinical Research Facility, Edinburgh. Libraries were prepared using the NEBNext Single Cell/Low Input RNA Library Prep Kit for Illumina according to the provided protocol. Paired-end sequencing was performed on the NextSeq 2000 platform (#20038897; Illumina Inc.)

using NextSeq 2000 P3 Reagents (200 Cycles). Basecall data were uploaded to BaseSpace and converted into fastq files.

Fastq files were downloaded from BaseSpace and analysed using protocols from the Data Carpentry workshop "RNAseq analysis with R," using self-generated code in the Edinburgh-based supercomputer Eddie, hosted at the University of Edinburgh High Performance Computing Cluster (Linux), via MobaXterm. A Fastqc module was used throughout for quality control of samples and to confirm adapter removal. Trim Galore was used to remove adapters (Illumina and GCTAATCATTGCAAGCAGTGGTATCAACGCAGAGTACAT). Burrows–Wheeler Aligner-MEM was used to align trimmed fastq files, creating sam files. SAMtools was used to convert files to bam files and sort by coordinates. HTSeq was used via anaconda to generate count tables for downstream analysis. The reference genome used was GRCz11, downloaded from Ensembl. Data were analysed and visualised using R. Data deposited at GSE311765.

Differential expression analysis was performed by the DESeq2 package (137). Significant genes were subset for those with an adjusted *P*-value of <0.05, and with a log$_2$ fold change. Data from DESeq analysis were transformed using the rlog function of the rld package before plotting. Annotation was provided by the AnnotationDbi package. Gene names were matched to Ensembl IDs using BioMart data from Ensembl. DEGs were determined to have *P* < 0.05 and a log$_2$ fold change of >1 or <−1 for an increase or reduction in enrichment, respectively.

## Zebrafish husbandry and analysis

All zebrafish embryos and adult fish were maintained in the Bioresearch and Veterinary Services (BVS) Aquatic Facility in the Queen's Medical Research Institute, the University of Edinburgh, in accordance with UK Home Office Regulations (Guidance on the Operations of Animals, Scientific Procedures Act, 1986). All protocols involving animal use obtained ethical approval from the University of Edinburgh. Experiments were undertaken under PPL PP1975958, PEE579666, PIL I86636769, and PIL I0890B33A. Day-to-day care was performed by trained animal technicians. Fish were maintained in a 14/10-h light/dark cycle with a water temperature of 28.5°C, pH between 7 and 7.5, and daily control of water quality, hardness, and conductivity. Embryos were obtained through natural matings, were not sorted by sex, and were staged referring to development at 28.5°C. Unless otherwise stated, all fish, including WT, had a WIK background. Adult fish were kept at approximately equal stocking densities throughout.

## Quantification and statistics

Parameters were quantified using FIJI/ImageJ (132) and tabulated with Microsoft Excel. For the screen, Crispant values were normalised to a gRNA-only control for each gene for each experimental repeat. Screen data were visualised and statistical analyses were performed using self-generated R code. RNAseq data visualisation was performed exclusively in R. General R coding was performed using ggplot2, tidyverse, ggrepel, extrafont, RColorBrewer, and viridis packages. Visualisation and statistical testing for the remaining experiments were performed in GraphPad Prism. Normality calculations were performed using D'Agostino and Pearson tests, or the Shapiro–Wilk

**Materials.**

| Reagent | Cat. #; Supplier |
|---------|------------------|
| 2X Brilliant III SYBR Green RT-qPCR Master Mix | 600882; Agilent |
| 6X Blue/Orange Loading Dye | G190A; Promega |
| Acrylamide (40%) | BP1408-1; Thermo Fisher Scientific |
| Agarose | A9539; Sigma-Aldrich |
| Ammonium persulphate (APS) | 10396503; Thermo Fisher Scientific |
| BcgI with NEB buffer 3.1 | R0545S; NEB |
| BD Microlance Needle (25G x 1"—Nr.18) | 300400; BD |
| BstXI with NEB buffer 3.1 | R0113S; NEB |
| Bsu36I with NEB buffer 1.1 (10X) | R0524S; NEB |
| Calcium nitrate tetrahydrate (Ca(NO$_3$)$_2$·4H$_2$O) | C1396; Sigma-Aldrich |
| Capillary glass (1.2 mm outer diameter, 0.68 mm inner diameter, 10.16 cm length, with filament) | 1B120F-4; World Precision Instruments |
| Cas9 nuclease | 1081059; IDT |
| Cell lysis buffer | 9803S; Cell Signaling Technologies |
| Chloroform | C2432; Honeywell |
| CRISPR constructs and primers | Various/IDT |
| DAPI | D9542; Sigma-Aldrich |
| DNA ladder | N0467S; NEB |
| E.Z.N.A. Plasmid DNA Mini Kit I | D6942-02; Omega Bio-Tek |
| EcoO109I with Neb CutSmart buffer | R0503S; NEB |
| Ethanol | E/0650DF/17; Thermo Fisher Scientific |
| Filter papers (Western blotting) | 88600; Thermo Fisher Scientific |
| Gel red | BT41003; Scientific Laboratory Supplies Ltd. |
| Glucose | G8270; Sigma-Aldrich |
| HEPES | H3375; Sigma-Aldrich |
| High-Capacity cDNA Reverse Transcription Kit | 4368814; Thermo Fisher Scientific |
| Hpy99I with NEB CutSmart buffer | R0615S; NEB |
| HRP2 antibodies (goat anti-rabbit) | P0448; DAKO |
| Immobilon Western Chemiluminescent HRP substrate | WBKLS0500; Millipore |
| Low melting point (LMP) agarose | 16500; Invitrogen |
| Magnesium sulphate (MgSO$_4$) | M2643; Sigma-Aldrich |
| Methyl cellulose | M0387; Merck Life Sciences |
| Microloader tips (0.5–20 μl) | F5242956003; SLS |
| NEB buffer 2 (10X) | B70025; NEB |
| NEBNext Single Cell/Low Input RNA Library Prep Kit | E6420; NEB |
| NextSeq 2000 P3 Reagents | 20040560; NEB |
| N-Phenylthiourea (PTU) | L06690.09; VWR |
| One*Taq* Hot Start DNA Polymerase | M0481G; NEB |
| PageRuler Prestained Protein Ladder | 26616; Thermo Fisher Scientific |
| PfoI with Tango buffer | 10568350; Thermo Fisher Scientific |
| Phenol red | P0290; Sigma-Aldrich |
| PhosSTOP tablets | 04906837001; Roche |
| Ponceau S solution | P7170; Sigma-Aldrich |

| Reagent | Cat. #; Supplier |
|---|---|
| Potassium chloride (KCl) | P9541; Sigma-Aldrich |
| PrimeSTAR Max Premix (2x) | R045; TaKaRa |
| Protease inhibitors | 4693124001; Sigma-Aldrich |
| RT-qPCR primers | Various; IDT |
| RNeasy Plus Micro Kit | 74034; QIAGEN |
| RNeasy Plus Mini Kit | 74136; QIAGEN |
| SacI with NEB CutSmart buffer (10X) | R0156S; NEB |
| Scalpel (No. 23) | 0510; Swann-Morton |
| SexAI with NEB rCutSmart buffer (10X) | R0605S; NEB |
| Sodium chloride (NaCl) | S7653; Sigma-Aldrich |
| Sodium dodecyl sulphate (SDS) | L3771; Sigma-Aldrich |
| Sodium hydroxide (NaOH) | S5881; Sigma-Aldrich |
| StrataClone PCR Cloning Kit | 240205; Agilent |
| T7 Endonuclease | M03025; NEB |
| Taz 1° Ab | cs8418; Cell Signaling Technologies |
| TEMED | 17919; Thermo Fisher Scientific |
| tracrRNA | 1073190; IDT |
| Triton X-100 | X100; Sigma-Aldrich |
| Trizma base | T6066; Sigma-Aldrich |
| TRIzol | 15596026; Ambion |
| Tryptone | 1279-7099; Thermo Fisher Scientific |
| Tween-20 | P1379; Sigma-Aldrich |
| UltraPure Distilled Water DNase/RNase Free | 10977-049; Invitrogen |
| XcmI with NEB buffer 2.1 | R0533S; NEB |
| X-gal | 16495; Cambridge BioScience |
| X-ray films | MOL7016; Scientific Laboratory Supplies |
| Yap1 1° Ab | cs4912; Cell Signaling Technologies |
| Yeast extract | 8013-01-2; Acros Organics |
| ZF imaging plate | HDK-ZFA101-02a; Funakoshi Co. |

test depending on sample size was used. The term "experimental repeats" used throughout is used to mean fish samples generated from a new injection day or breeding, and an independent experiment is performed as described before.

# Data Availability

RNAseq data are deposited at GSE311765.

# Supplementary Information

# Acknowledgements

SE Riley was funded by Wellcome Trust PhD Studentship (108906/Z/15/Z). M Noskova Fairley was funded by a MRC Precision Medicine DTP Studentship. Y Feng was funded by Wellcome Trust sir Henry Dale Fellowship (100104/Z/12/Z) and Cancer Research UK Early Detection Award (C38363/A26931). Ongoing research in the Hansen laboratory (CG Hansen) was funded by Worldwide Cancer Research (19-0238) and CSO-LifeArc. This project was initiated by pump prime funding from ISSF3. Professor B Link is acknowledged for sharing the *ctgfa*:dGFP construct in order for us to rederive the transgenic reporter fish. We thank the University of Edinburgh BVS Aquatics Unit for housing the fish and the CALM and IRR microscopy facilities for excellent support with imaging. Dr. R Wiegand, QMRI CALM imaging facility, is acknowledged with help setting up the post-imaging processing of spectral unmixing for the HCR RNA-FISH protocol. BioRender is acknowledged for software to generate schematics. For the purpose of open access, the author has applied a Creative Commons

## Author Contributions

SE Riley: data curation, formal analysis, investigation, visualisation, methodology, and writing—original draft, review, and editing.
M Noskova Fairley: formal analysis and investigation.
S Xia: formal analysis and investigation.
R Cunningham: data curation, investigation, and writing—review and editing.
J Cholewa-Waclaw: investigation.
Y Feng: resources, supervision, funding acquisition, methodology, project administration, and writing—review and editing.
CG Hansen: conceptualisation, resources, formal analysis, supervision, funding acquisition, investigation, visualisation, methodology, project administration, and writing—original draft, review, and editing.

## Conflict of Interest Statement

The authors declare that they have no conflict of interest.

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
