## [Reviewer comments · Life Science Alliance]

In vivo screen reveals specific roles of Hippo pathway components in development and regeneration

Susanna Riley, Michaela Fairley, Shijia Xia, Richard Cunningham, Justyna Cholewa-Waclaw, Yi Feng, and Carsten Hansen
DOI: <https://doi.org/10.26508/lsa.202503296>

Corresponding author(s): Carsten Hansen, University of Edinburgh

Review Timeline:

Submission Date:	2025-03-06
Editorial Decision:	2025-04-23
Revision Received:	2025-10-21
Editorial Decision:	2025-11-17
Revision Received:	2025-12-01
Accepted:	2025-12-04

Scientific Editor: Sarita Hebbar

Transaction Report:

April 23, 2025

Re: Life Science Alliance manuscript #LSA-2025-03296-T

Dr. Carsten G Hansen
University of Edinburgh
Center for Inflammation Research
Institute for Regeneration and Repair
Edinburgh BioQuarter, 4-5 Little France Drive
Edinburg EH16 4UU
United Kingdom

Dear Dr. Hansen,

Thank you for submitting your manuscript entitled "In vivo screen reveals specific roles of Hippo pathway components in development and regeneration" to Life Science Alliance. The manuscript was assessed by three expert reviewers, whose comments are appended to this letter.

Overall, the reviewers find this work is of potential value to the community. That said, we agree with the reviewers that several significant points in the manuscript must be addressed for publication at LSA. These are:

1. Expand on existing data/new experimental data to be included:
 - A. Rescue experiments as suggested by Reviewer 1, point 3.
 - B. Provide efficiency and specificity data for gene editing:
 - i. Include off-target analysis or control injections (reviewer 1, point 1 and reviewer 3, point 2)
 - ii. Include tests for efficiency and protein levels completed for the remaining 5 genes (Reviewer 1, point 1 and Reviewer 3, point 3). Further, specific if efficiency and/or protein levels for every CRISPR (to validate that each injection was successful) was performed, or only once for each pair of gRNAs. Also specify (for all crisprants) if phenotyped animals are carry the expected mutations (Reviewer 3, point 7).
 - C. Provide additional characterisation of the new reporter line, in normal developing embryos and to assess if all expression domains are affected in the stable yap1 mutant line (Reviewer 2, final section)
2. Sections of the manuscript must be revised/restructured with changes to the text and inclusion of tables as suggested by the reviewers.

In view of these recommendations, we invite you to submit a revised manuscript addressing the reviewers' comments. When submitting the revision, please include a letter addressing all the reviewers' comments point by point. While a rebuttal must respond to all points in some form, additional data to resolve these points (other than ones indicated above) is not required.

To upload the revised version of your manuscript, please log in to your account: <https://lsa.msubmit.net/cgi-bin/main.plex>
While you are revising your manuscript, please also attend to the below editorial points to help expedite the publication of your manuscript. Please direct any editorial questions to the journal office.

Thank you for this interesting contribution to Life Science Alliance. We are looking forward to receiving your revised manuscript.

Sincerely,

Sarita Hebbar, PhD
Scientific Editor
Life Science Alliance
<http://www.lsjournal.org>

- A letter addressing the reviewers' comments point by point.
- An editable version of the final text (.DOC or .DOCX) is needed for copyediting (no PDFs).
- High-resolution figure, supplementary figure and video files uploaded as individual files: See our detailed guidelines for preparing your production-ready images, <https://www.life-science-alliance.org/authors>
- Summary blurb (enter in submission system): A short text summarizing in a single sentence the study (max. 200 characters including spaces). This text is used in conjunction with the titles of papers, hence should be informative and complementary to the title and running title. It should describe the context and significance of the findings for a general readership; it should be written in the present tense and refer to the work in the third person. Author names should not be mentioned.
- By submitting a revision, you attest that you are aware of our payment policies found here: <https://www.life-science-alliance.org/copyright-license-fee>

B. MANUSCRIPT ORGANIZATION AND FORMATTING:

Reviewer #1 (Comments to the Authors (Required)):

This study uses zebrafish larvae and a high-efficiency CRISPR/Cas9 F0 knockout approach to investigate the roles of individual Hippo pathway components including Yap1, Taz (Wwtr1), and their upstream regulators during vertebrate development and regeneration. By targeting single and paired paralogs, the authors uncover both redundant and distinct functions of Hippo pathway genes. Two regenerative paradigms (mild and severe tail fin injury) reveal that Yap1 and Taz play differential roles depending on injury context. Notably, a *ctgfa:d2GFP* reporter shows dynamic and non-redundant Yap/Taz activity during regeneration.

The manuscript provides significant progress in the role of Hippo pathway that had not been accessed in previous reports.

Here are several comments that could strengthen the manuscript.

1. The study employs CRISPR/Cas9-mediated F0 knockouts in zebrafish to dissect the roles of Hippo pathway components in development and regeneration. However, the efficiency and specificity of gene editing for some targets (e.g., *ctgfa/b*, *nf2a/b*) could be clarified further. While restriction digest and western blotting are shown for Yap1 and Taz, it would strengthen the paper if similar validation (e.g., RT-qPCR, western blot, or T7 endonuclease assay) were provided for other key components—particularly where overlapping or compensatory functions are inferred. Including off-target analysis or control injections would help address concerns of non-specific effects.

2. The authors employ both mild and severe tail fin injury paradigms, which is indeed a strength of the study. However, the description of how injury extent is controlled and measured could be more clearly defined.

3. The manuscript highlights functional divergence and redundancy among Hippo components, particularly Yap1 and Wwtr1/Taz. However, the conclusion that Yap and Taz serve non-redundant roles in specific regenerative contexts could be further substantiated. Could the authors consider performing rescue experiments (e.g., mRNA injection of *yap1* or *wwtr1* in their respective Crispants)? This would strengthen the claim of functional non-redundancy, particularly since previous studies have reported overlapping activity in early development.

4. While the zebrafish model is well suited for in vivo genetic screening, the translational relevance to mammalian development and regeneration could be discussed more explicitly. Since the authors highlight potential implications for regenerative therapies and Hippo-targeting compounds, it would be useful to align key findings (e.g., differential roles of Yap and Taz) with known mammalian phenotypes or pathologies. This would help position the zebrafish findings within a broader vertebrate context and enhance the clinical relevance of the work.

Reviewer #2 (Comments to the Authors (Required)):

The authors have knocked down a number of components of the Hippo pathway in zebrafish embryos using CRISPR-Cas9 technology and analyzed phenotypes seen in the embryos injected with the gRNA+Cas9 ("Crispants"). They provide quantification of several parameters of overall morphology of developing embryos (length, eye size) and studied regeneration of the tail of the embryos after two types of injuries; one removing the fin, the other cutting additionally into the notochord and neural tube. Regeneration is largely assessed by quantification of the amount of tissue that has regrown, but also using bulk transcriptomics of wild-types and mutants. In addition to analysing Crispants, the authors established one stable mutant line of yap1. Furthermore, they establish a reporter transgenic line for Yap/Taz signaling.

I appreciate the author's argument that a systematic comparison between mutants in several components of the Hippo pathway in the same system is useful, despite the relatively large body of knowledge that has already been acquiring regarding the roles of this pathway in growth and development, and also in regeneration. I also acknowledge that the authors have invested a large amount of work that went into creating and analyzing Crispants of 9 genes, plus several dual knockdowns of two genes at once. The transcriptomics data they have created is also a useful resource for future studies. The data are generally conclusive, and the correct quantifications and statistical analyses have been performed. Also, the methods provide enough details.

The problem with the paper is that it does not really derive in interesting novel biological insight. There are some potentially interesting differences in phenotypes between paralogs of the same gene or between yap and taz, also between the requirement of several genes for the mild vs. more severe tail injury. However, if one would like to get serious about understanding these potentially interesting differences in function of the Hippo pathway components, one would have to - in my opinion - analyze well characterized null mutants. The variability inherent to the Crispant approach, were a mixture of different genetic lesions are present in different cells and embryos, makes it in my opinion impossible to derive unambiguous strong conclusions about the central question here, which is a careful side-by-side comparison of loss-of-function of the diverse Hippo pathway components. The authors themselves observe that the phenotypes observed in the yap1 stable mutant vs the Crispants are not exactly the same.

Overall, I thus fear that only careful side-by-side analysis of stable null mutants will provide the additional insight beyond the current state of the literature that the authors are seeking. Whether the large collection of data that is presented here in the absence of much novel insight into the function of Hippo signaling in development or regeneration is of sufficient interest for readers of Life Sci Alliance is for the editors to decide.

If the authors are asked to revise, the only strengthening of existing data that I would recommend is to further show that the new reporter line is indeed reporting Yap/Taz signaling. The authors only show some quantifications of signal in mutants after tail injuries. What about the signal seen in normal developing embryos? Are all expression domains affected in their stable yap1 mutant line?

Reviewer #3 (Comments to the Authors (Required)):

The manuscript, "In vivo screen reveals specific roles of Hippo pathway components in development and regeneration," by Riley et al., begins to provide insights into the roles of different Hippo pathway genes in larval development and fin fold regeneration. They use primarily F0 CRISPR screens to follow the phenotypes of crispants for 7 components of the Hippo pathway. While several phenotypes are cataloged for all 7 components, as written it is hard to contextualize the relevance of the impacts. There are also some concerns about whether the individual larval phenotypes reflect specific genetic changes in the predicted gene. Overall, this manuscript includes a wide range of data, but does not fully explain the experimental design so there are some questions about the reliability of the conclusions. Also, while there is a lot of data, much of it is examined superficially, which limits its impact. Below I identify some specific concerns and provide some suggestions for the authors.

1. It is not clear why morpholinos, chemical inhibitors, or stable mutant lines "do not allow for direct comparative analysis across multiple Hippo pathway components." (Introduction).

2. Regarding the CRISPR screen, the authors should comment on the possibility of off-target mutations possibly causing phenotypes. Also, the authors should say if they monitored efficiency and/or protein levels every time they performed CRISPR (to validate that each injection was successful), or only once for each pair of gRNAs.

3. It sounds like the authors demonstrated that the CRISPR approach was efficient for targeting the yap1 and wwtr1 genes, and then used the same approach for the remaining 5 genes. Were similar tests for efficiency and protein levels completed for the remaining 5 genes (i.e. since each genetic target is independent, the efficiency of gene knockout should be tested for all targets)? If so, this should be stated in the Results. If not, the authors should explain why this was not completed.

4. Can the authors provide any insights into why there would be differential phenotypes for the Hippo pathway genes (i.e. especially the signaling components)? Are the selected proteins known to be involved in multiple pathways? Is it possible that some of the observed phenotypes are due to off-target effects?

5. Since the mild regeneration paradigm is believed to be regulated mainly by changes in cell proliferation, it would be nice to also show differential cell division in the crispants with reduced or enhanced regeneration.
6. Overall, it is hard to contextualize the relevance of the impacts, and hard to remember the difference in phenotypes between different genes. Perhaps the authors could include a table of the different genes with the phenotypes under the different conditions. How do these phenotypes compare to what is already known? Which findings are novel?
7. For the yap1 crispants which seem not to show a severe regeneration effect (pages 5-6), are the authors suggesting that the reason they did not see an effect is because most of the actual crispants died prior to the analyses (and so they were actually evaluating individuals without mutations)? If so, this raises serious concerns about the analyses of all of the data. How do the authors know that the phenotypes they observe are due to the mutations in the targeted genes? Related, I suggest completed rewriting these sections to bring the stable mutation data proximal to the crispant data. As is, the authors make a conclusion about the yap1 crispants that they later alter based on the stable line. And, the authors should include an explanation (for all crispants) of how they know the animals they are evaluating carry the expected mutations (vs. being unedited survivors in the population).
8. The analyses of the notochord regeneration and the transcriptomics are interesting, but very superficial and rushed. I suggested further exploring the notochord regeneration here, and expanding the transcriptomics into a separate manuscript where hypotheses can be proposed, and mechanisms can be more deeply explored and tested.
9. The text would benefit from careful editing for clarity. The Results are very long and sometimes redundant, and it can be difficult to identify the most relevant information from each section.
10. For a manuscript with 7 figures and 7 supplemental figures, the Discussion is very brief, and does not connect the findings to the bigger picture. Some ideas for how the authors might help the reader to understand the significance of the results include: explain the "specific roles" of each pathways component as stated in the article title; compare crispant phenotypes to known stable mutants, or to morphants; explain the next important questions or ideas for future experiments.

Dear editor and reviewers

We thank the editor and reviewers for their time, and for the overall positive and constructive feedback. We appreciate the patience while we have readied the updated manuscript. We have had to prioritize experiments due to current team composition. We have added substantial additional data, including on Yap and Taz involvement in macrophage subgroup localisation to the regenerative tail fin (**New Fig 8 + Sup Fig 8**), rewritten sections for clarity, restructured the storyline including moving previous Fig 7 to current Fig 6, as well as implemented feedback throughout. We believe that this implementation has greatly improved the manuscript. We are hopeful that the manuscript is now ready for acceptance.

We thank you for your time and feedback.

Our response in red

EDITOR COMMENTS

Overall, the reviewers find this work is of potential value to the community. That said, we agree with the reviewers that several significant points in the manuscript must be addressed for publication at LSA. These are:

1. Expand on existing data/new experimental data to be included:

A. Rescue experiments as suggested by Reviewer 1, point 3.

While rescue experiments in theory confirm on target specificity, rescue experiments, especially in a developing zebrafish are nontrivial. It is often critical to only rescue the expression to endogenous levels, as overexpression frequently give rise to spurious phenotypes. Additional complications might arise from mosaic expression and temporal mismatch. We have therefore not prioritized this experiment.

B. Provide efficiency and specificity data for gene editing:

i. Include off-target analysis or control injections (reviewer 1, point 1 and reviewer 3, point 2)

All Crispant data are shown normalised to their gRNA-only control.

Cas9-only injected control and uninjected embryos were also analysed, data included for **Sup Fig 2B, C** as an example. The CRISPR guides were designed to minimise potential off-target effects. The impact of any infrequent off-target effects is minimised due to the nature of the Crispant approach, where each Crispant represents one data point, and any potential rare nonspecific events will therefore only be present in that individual larvae. Potential infrequent off-target effects are consequently unlikely to have an impact on the overall analysis and is a likely benefit from the Crispant approach. We have included further text on this in the discussion section.

ii. Include tests for efficiency and protein levels completed for the remaining 5 genes (Reviewer 1, point 1 and Reviewer 3, point 3).

The appropriate antibodies to recognise fish proteins are not available. Antibodies are commonly designed to recognise human epitopes. We evaluated and confirmed specificity of the Yap1 and Taz antibodies, which confirmed that our Crispant approach works. We have added context in the discussion, that we cannot rule out a full loss of protein in all larvae tested, but that the impact from potential spurious non-productive events will be limited in the data analysis, due to the nature of the single larvae Crispant approach. We have also added further information elsewhere.

Further, specific if efficiency and/or protein levels for every CRISPR (to validate that each injection was successful) was performed, or only once for each pair of gRNAs.

PCR/RE digest analysis was performed once during the guide design process in multiple individual embryos for all guides (except *yap1*, where guide design hindered analysis of individual embryos, and where protein analysis shows loss of Yap protein, (**Sup Fig 1 & Fig 1**)). Guides were chosen where analyses showed 100% guide mutagenesis efficiency (**Sup Fig 1**). Analysis of each injection was not performed – as this was outside the feasibility for the screening process. Notably, different batches of Crispants show similar phenotypic spread.

Also specify (for all crispants) if phenotyped animals are carry the expected mutations (Reviewer 3, point 7).

We show almost complete loss of protein in pooled *yap1* Crispants and *wwtr1* Crispants (**Fig 1**), highlighting that our approach is effective. We acknowledge that we don't fully know that each embryo carries the expected loss of function mutations, but we do show that CRISPR activity (efficient guide mutagenesis) is present in close to 100% of Crispant larvae (**Sup Fig 1**). The two gene-specific guides have been designed to have functional consequences. We include text on this in the discussion. The use of large numbers of embryos per experiment and three independent experimental day repeats is expected to reduce the effect of any small proportion of non-mutated embryos.

C. Provide additional characterisation of the new reporter line, in normal developing embryos and to assess if all expression domains are affected in the stable yap1 mutant line (Reviewer 2, final section)

This is not a new reporter line, we apologise if we didn't make this clear. It has been published elsewhere and characterised well (Miesfeld et al 2015 and Miesfeld and Link, 2014) (ref 5 and 6 in the manuscript). We took advantage of the Tol2-ctgf(-1kb)-d2GFP-pA reporter plasmid (provided by Prof B Link) and co-injected with Tol2 mRNA at the 1 cell stage. Fish were then outcrossed, and later incrossed to homozygosity, as described in the materials and methods. Our expression data replicate well with what was reported by the Link lab. **Sup Fig 7** show stills of videos of confocal z stacks taken through a normal embryo over the first 5 days of development, and we now also include the videos of these image stacks

The stable *yap1* mutant line struggles to breed. Please note the low survival rate (**Sup 4D**) for context. We have therefore not been able to generate a mutant line in the reporter line background to assess any effects on expression domains.

2. Sections of the manuscript must be revised/restructured with changes to the text and inclusion of tables as suggested by the reviewers.

We have updated the text to incorporate the feedback, restructured the manuscript, expanded on the discussion section and included more reference to provide context. We have also included a suggested table, please see heatmap in (new) **Fig 2F**.

In view of these recommendations, we invite you to submit a revised manuscript addressing the reviewers' comments. When submitting the revision, please include a letter addressing all the reviewers' comments point by point. While a rebuttal must respond to all points in some form, additional data to resolve these points (other than ones indicated above) is not required.

Thank you for this interesting contribution to Life Science Alliance. We are looking forward to receiving your revised manuscript.

Sincerely,

Sarita Hebbar, PhD
Scientific Editor
Life Science Alliance
<http://www.lsjournal.org>

Reviewer #1 (Comments to the Authors (Required)):

This study uses zebrafish larvae and a high-efficiency CRISPR/Cas9 F0 knockout approach to investigate the roles of individual Hippo pathway components including Yap1, Taz (Wwtr1), and their upstream regulators during vertebrate development and regeneration. By targeting single and paired paralogs, the authors uncover both redundant and distinct functions of Hippo pathway genes. Two regenerative paradigms (mild and severe tail fin injury) reveal that Yap1 and Taz play differential roles depending on injury context. Notably, a *ctgfa:d2GFP* reporter shows dynamic and non-redundant Yap/Taz activity during regeneration.

The manuscript provides significant progress in the role of Hippo pathway that had not been accessed in previous reports.

We thank the reviewer for his/her positive feedback, and we are pleased that the reviewer shares our positive evaluation of the impact of this study.

Here are several comments that could strengthen the manuscript.

1. The study employs CRISPR/Cas9-mediated F0 knockouts in zebrafish to dissect the roles of Hippo pathway components in development and regeneration. However, the efficiency and specificity of gene editing for some targets (e.g., *ctgfa/b*, *nf2a/b*) could be clarified further. While restriction digest and western blotting are shown for Yap1 and Taz, it would strengthen the paper if similar validation (e.g., RT-qPCR, western blot, or T7 endonuclease assay) were provided for other key components-particularly where overlapping or compensatory functions are inferred. Including off-target analysis or control injections would help address concerns of non-specific effects.

Restriction digest of each guide is shown in **Sup Fig 1**. As described in the compiled general comments highlighted by the editor, Western blotting could not be performed in general for genes other than *yap1* or (*wwtr1*)Taz, primarily as antibodies are not commercially available (unfortunately the zebrafish field has a general lack of appropriate antibodies). Control injections are included – uninjected and Cas9-only injections were performed (e.g. **Fig 3B**, **Sup Fig 2**), though data is not shown for all assays. All screening results in **Figs 2 and 3** are shown as Crispant values normalised to individual gRNA-only controls performed alongside the Crispant analysis to reduce inter-batch variability and to reduce the effect of any gRNA-mediated effects.

2. The authors employ both mild and severe tail fin injury paradigms, which is indeed a strength of the study. However, the description of how injury extent is controlled and measured could be more clearly defined.

Thanks for the feedback. In addition to the description in **Fig 3A** and supplementary figure 2A, we have also clarified this further in the text.

3. The manuscript highlights functional divergence and redundancy among Hippo components, particularly Yap1 and *Wwtr1/Taz*. However, the conclusion that Yap and Taz serve non-redundant roles in specific regenerative contexts could be further substantiated. Could the authors consider performing rescue experiments (e.g., mRNA injection of *yap1* or *wwtr1* in their respective Crispants)? This would strengthen the claim of functional non-redundancy, particularly since previous studies have reported overlapping activity in early development.

As highlighted in our comments to the condensed reviewer and editor feedback compiled by the editor, we didn't attempt this, as we prioritized our efforts elsewhere, especially due to the current team structure.

4. While the zebrafish model is well suited for in vivo genetic screening, the translational relevance to mammalian development and regeneration could be discussed more explicitly. Since the authors highlight potential implications for regenerative therapies and Hippo-targeting compounds, it would be useful to align key findings (e.g., differential roles of Yap and Taz) with known mammalian phenotypes or pathologies. This

would help position the zebrafish findings within a broader vertebrate context and enhance the clinical relevance of the work.

This is a good point.

We now include additional discussions in the text and references for this important topic. We agree that this provides additional context for our studies and further encourage general interest.

Reviewer #2 (Comments to the Authors (Required)):

The authors have knocked down a number of components of the Hippo pathway in zebrafish embryos using CRISPR-Cas9 technology and analyzed phenotypes seen in the embryos injected with the gRNA+Cas9 ("Crispant"). They provide quantification of several parameters of overall morphology of developing embryos (length, eye size) and studied regeneration of the tail of the embryos after two types of injuries; one removing the fin, the other cutting additionally into the notochord and neural tube. Regeneration is largely assessed by quantification of the amount of tissue that has regrown, but also using bulk transcriptomics of wild-types and mutants. In addition to analysing Crispants, the authors established one stable mutant line of yap1. Furthermore, they establish a reporter transgenic line for Yap/Taz signaling.

I appreciate the author's argument that a systematic comparison between mutants in several components of the Hippo pathway in the same system is useful, despite the relatively large body of knowledge that has already been acquiring regarding the roles of this pathway in growth and development, and also in regeneration. I also acknowledge that the authors have invested a large amount of work that went into creating and analyzing Crispants of 9 genes, plus several dual knockdowns of two genes at once. The transcriptomics data they have created is also a useful resource for future studies. The data are generally conclusive, and the correct quantifications and statistical analyses have been performed. Also, the methods provide enough details.

The problem with the paper is that it does not really derive in interesting novel biological insight. There are some potentially interesting differences in phenotypes between paralogs of the same gene or between yap and taz, also between the requirement of several genes for the mild vs. more severe tail injury. However, if one would like to get serious about understanding these potentially interesting differences in function of the Hippo pathway components, one would have to - in my opinion - analyze well characterized null mutants. The variability inherent to the Crispant approach, where a mixture of different genetic lesions are present in different cells and embryos, makes it in my opinion impossible to derive unambiguous strong conclusions about the central question here, which is a careful side-by-side comparison of loss-of-function of the diverse Hippo pathway components. The authors themselves observe that the phenotypes observed in the yap1 stable mutant vs the Crispants are not exactly the same.

We appreciate the inherent variability in the Crispant approach creating different genetic lesions. However, to us this means that significant phenotypes are independent of one specific lesion and can instead be induced by a range of different mutations and may

therefore be a consistent phenotype. Importantly, the approach for Yap and Taz show an almost total loss of protein for pooled embryos (**Fig 1**).

A further strength of the zebrafish embryo CRISPR system is that the CRISPR can be delivered at the one cell stage, so reducing the likelihood of different genetic lesions in different cells.

Overall, I thus fear that only careful side-by-side analysis of stable null mutants will provide the additional insight beyond the current state of the literature that the authors are seeking. Whether the large collection of data that is presented here in the absence of much novel insight into the function of Hippo signaling in development or regeneration is of sufficient interest for readers of Life Sci Alliance is for the editors to decide.

Our studies are the first side by side *in vivo* comparison of loss of function components of the Hippo pathway in the vertebrates. While acknowledging that no model is perfect, we argue, that this approach and the findings are of general interest for a wide readership. We include further text in the discussion section on this aspect. Additionally, we now include further new biological insights, obtained from analysis of macrophage function in *yap1* and *wvtr1* Crispants.

If the authors are asked to revise, the only strengthening of existing data that I would recommend is to further show that the new reporter line is indeed reporting Yap/Taz signaling. The authors only show some quantifications of signal in mutants after tail injuries. What about the signal seen in normal developing embryos?

The approach was established by the Link lab. Our expression data confirms their findings. We have included clarification above on this reporter line in the condensed editor feedback and in the text.

Are all expression domains affected in their stable *yap1* mutant line?

We provide in **Sup Fig 4** an overview of the sequencing and translated peptide (**Sup Fig 4C**), highlighting the loss of Yap protein.

We thank the reviewer for his/her feedback.

Reviewer #3 (Comments to the Authors (Required)):

The manuscript, "In vivo screen reveals specific roles of Hippo pathway components in development and regeneration," by Riley et al., begins to provide insights into the roles of different Hippo pathway genes in larval development and fin fold regeneration. They use primarily F0 CRISPR screens to follow the phenotypes of crispants for 7 components of the Hippo pathway. While several phenotypes are cataloged for all 7 components, as written it is hard to contextualize the relevance of the impacts. There are also some concerns about whether the individual larval phenotypes reflect specific genetic changes in the predicted gene. Overall, this manuscript includes a wide range of data, but does not fully explain the

experimental design so there are some questions about the reliability of the conclusions. Also, while there is a lot of data, much of it is examined superficially, which limits its impact. Below I identify some specific concerns and provide some suggestions for the authors.

1. It is not clear why morpholinos, chemical inhibitors, or stable mutant lines "do not allow for direct comparative analysis across multiple Hippo pathway components." (Introduction).

Thanks for the comment.

Targeting non-enzymatic proteins is challenging with small molecule approaches. Chemical inhibitors are not available for most Hippo components. The development of Hippo pathway targeting compounds is in their infancy. Historically, the lack of component specific Hippo pathway targeting drugs have made research into the Hippo pathway challenging, and studies such as ours must rely on genetic targeting. Although the potential use of such compounds, if component specific in the fish, would allow for appropriate analysis, technical limitations on the development of these make such comparisons impossible. The focus has been on inhibiting YAP/TAZ-TEAD for cancer biology (Cunningham and Hansen, 2023, Cunningham et al., 2025). As this transcriptional YAP/TAZ-TEAD complex does not have enzymatic activity, even the development of specific inhibitors of these prominent Hippo pathway members have been challenging. The specificity in fish in general, and especially for paralogues is currently unknown.

Stable mutant lines could be generated, and as highlighted here, we generate these for *yap1*, showing that the Crispants overall phenocopy their stable mutants. We discuss the observed discrepancies. Generating stable mutants for the full set of analysed genes is outside the scope of the current study.

Morpholinos are powerful research tools, but have some specificity concerns, including p53 activation, divergent target gene concentrations, off target effects and additional don't provide full null animals (Stainer et al., 2017, doi: 10.1371/journal.pgen.1007000). In essence, no approach, chemical, morpholino or genetic targeting is perfect.

We have added further text covering this important aspect in the discussion.

2. Regarding the CRISPR screen, the authors should comment on the possibility of off-target mutations possibly causing phenotypes. Also, the authors should say if they monitored efficiency and/or protein levels every time they performed CRISPR (to validate that each injection was successful), or only once for each pair of gRNAs.

We thank the reviewer for their comment. We agree this is an important aspect of any screen. We have discussed this above

*3. It sounds like the authors demonstrated that the CRISPR approach was efficient for targeting the *yap1* and *wwtr1* genes, and then used the same approach for the remaining 5 genes. Were similar tests for efficiency and protein levels completed for the remaining 5 genes (i.e. since each genetic target is independent, the efficiency of gene knockout should*

be tested for all targets)? If so, this should be stated in the Results. If not, the authors should explain why this was not completed.

We thank the reviewer for their comment.

This is discussed above, including highlighting limited reagents working in zebrafish.

4. Can the authors provide any insights into why there would be differential phenotypes for the Hippo pathway genes (i.e. especially the signaling components)? Are the selected proteins known to be involved in multiple pathways? Is it possible that some of the observed phenotypes are due to off-target effects?

This aspect is interesting, but the answer complex. It is likely that both different components and paralogues are differentially expressed, both temporarily and in different cell types. It is also likely that NF2 loss only in some instances will have a prominent effect on the Hippo pathway, where loss of LATS1/2 are likely detrimental to most of YAP/TAZ regulation. We recently, using an isogenic mammalian cellular model identified NF2 as a mechano-transducer component within the Hippo pathway; where NF2 loss in stiff environments had little effects on YAP, but in soft cellular environments led to YAP hyperactivation (Cunningham et al., 2023, 2025 (ref 70 and 124 in the manuscript)). Some of the proteins, such as the YAP/TAZ-TEAD target genes *CYR61* and *CTGF* furthermore encode secreted proteins with multiple effects. Determining these detailed molecular spatio-temporal mechanisms are certainly of clear interest, but resolving these complexities are for future studies.

5. Since the mild regeneration paradigm is believed to be regulated mainly by changes in cell proliferation, it would be nice to also show differential cell division in the crispants with reduced or enhanced regeneration.

We agree that this would potentially be interesting, however this would require substantial imaging, which is currently prohibitive to us due to costs and the composition of the current research team. We have therefore been unable to conduct this experiment.

6. Overall, it is hard to contextualize the relevance of the impacts, and hard to remember the difference in phenotypes between different genes. Perhaps the authors could include a table of the different genes with the phenotypes under the different conditions. How do these phenotypes compare to what is already known? Which findings are novel?

We thank the reviewer for his/her comment. We have now included additional text, further references and added phenotype summary heatmaps (**Fig 2F**) for context.

7. For the yap1 crispants which seem not to show a severe regeneration effect (pages 5-6), are the authors suggesting that the reason they did not see an effect is because most of the actual crispants died prior to the analyses (and so they were actually evaluating individuals without mutations)? If so, this raises serious concerns about the analyses of all of the data. How do the authors know that the phenotypes they observe are due to the mutations in the targeted genes? Related, I suggest completed rewriting these

sections to bring the stable mutation data proximal to the crispant data. As is, the authors make a conclusion about the *yap1* crispants that they later alter based on the stable line. And, the authors should include an explanation (for all crispants) of how they know the animals they are evaluating carry the expected mutations (vs. being unedited survivors in the population).

Thanks for the comment.

It is a valid point. We were initially puzzled ourselves, which led us to explore this early on. Please see attached data, where we separate out "severe" and "normal" phenotypic larvae. Both the severe disrupted phenotypic *yap1* Crispants and the "normal" phenotypic *yap1* Crispants have lost most of Yap protein (C).

Yap1 Crispant

A) Representative images of WT embryos and "normal" (2 larvae) and "disrupted" phenotype *yap1* Crispants. **B)** DNA gel showing a T7E1 endonuclease assay of controls and *yap1* Crispant embryos, separated into "normal" and "disrupted" phenotypes. Both Crispant groups show the same level of DNA mutagenesis. U=undigested sample; D=digested sample (with T7E1). Arrows show band shift indicative of efficient *yap1* mutagenesis. **C)** Yap Western blot of controls and *yap1* Crispant embryo, separated into "normal" and "disrupted" phenotypes, and a "pooled" sample with no separation. There was little difference between Crispant groups in residual Yap protein levels. Ponceau stain as loading control.

Cas9=Cas9-only injected fish; gRNA=*yap1* CRISPR guide RNA-only injected fish; CRISPrant=*yap1* CRISPR mutagenized fish.

We hypothesise, that during development Yap is necessary to favour a normal development at a specific phase, and that if this development occurs normally anyhow,

then Yap is less critical for the later part of development. Currently this is a working hypothesis. We don't have conclusive experimental data yet.

We have kept the ordering of the Crispant and the stable mutant as is. We have restructured other parts of the manuscript to improve the flow (including changing the order of Fig 6 and Fig 7).

8. The analyses of the notochord regeneration and the transcriptomics are interesting, but very superficial and rushed. I suggested further exploring the notochord regeneration here, and expanding the transcriptomics into a separate manuscript where hypotheses can be proposed, and mechanisms can be more deeply explored and tested.

We thank the reviewer for the suggestion. We have opted to include the data here, as we feel it is an important part and that the datasets are useful to the field and for the overall impact of the manuscript. We have now also expanded our analysis of a potential mechanisms with a focus on macrophages in the severe regenerative paradigm (**Fig 8** and **Sup Fig 8**).

9. The text would benefit from careful editing for clarity. The Results are very long and sometimes redundant, and it can be difficult to identify the most relevant information from each section.

We thank the reviewer for his/her feedback. We have now edited and restructured the manuscript, so for example the *ctgfa* reporter comes before the transcriptional analysis, which then links to the macrophage involvement. We have also included a longer discussion section. We agree this section was short in the previous version.

10. For a manuscript with 7 figures and 7 supplemental figures, the Discussion is very brief and does not connect the findings to the bigger picture. Some ideas for how the authors might help the reader to understand the significance of the results include: explain the "specific roles" of each pathways component as stated in the article title; compare crispant phenotypes to known stable mutants, or to morphants; explain the next important questions or ideas for future experiments.

We agree that the previous discussion section was too short. We have now expanded the discussion and edited the text to highlight the context of the findings and discuss the challenges and strengths of the Crispant screening approach. We also discuss future directions.

We thank the reviewer for his/her time and insightful feedback.

Overall, we thank the reviewers for their time and feedback. We are hopeful that our manuscript is now ready for publication.

November 17, 2025

RE: Life Science Alliance Manuscript #LSA-2025-03296-TR

Dr. Carsten Gram Hansen
University of Edinburgh
Center for Inflammation Research
Institute for Regeneration and Repair
Edinburgh BioQuarter, 4-5 Little France Drive
Edinburg EH16 4UU
United Kingdom

Dear Dr. Hansen,

Thank you for submitting your revised manuscript entitled "In vivo screen reveals specific roles of Hippo pathway components in development and regeneration". Your revised manuscript was evaluated by all the original reviewers whose comments are appended below. As you will read, the three reviewers are consistent in their views that the revised manuscript has addressed all their previous concerns.

In line with the reviewers' evaluation, we would be happy to publish your paper in Life Science Alliance pending final revisions necessary to meet our formatting guidelines.

- Kindly confirm if following images are different images: (a) Figure 3B, WT and Figure S5A, WT, (b) Figure 3B, yap1 Crispant and Figure S5A, yap1 Crispant, (c) Figure S5A, WT and Figure S3 WT, (d) Figure S5A, yap1 Crispant and Figure S3, yap1 Crispant. If the same image has been utilised in multiple panels, then please indicate as such in the figure legends of all the panels.
- Thank you for providing a detailed list of all used materials in the table named as 'Materials'. We encourage you to make a general reference to this table at the start of the methods section.
- Please include details of objectives (Name, numerical aperture) and the temperature for live-imaging in the methods.
- We encourage you to change the title of one of the methods sub-sections called 'Opera Imaging'. We suggest 'Whole body imaging for macrophage sub-types' as an option.
- Please provide the details for reagents and methods for RNA-FISH (Supplementary Figure 6).
- For Western Blot quantification, please clarify if loading values were determined by use of a loading control as the materials table mentioned the use of "Gapdh 1 Ab". This sentence in the methods, "Loading levels were determined by analysing the peak height between ~55-100 kDa (where possible) of a Ponceau blot." refers to another approach for obtaining loading levels. Please clarify this sentence and the use of Gapdh 1 Ab.
- Kindly include details of the secondary antibodies in the table.
- Please upload your main manuscript text as an editable doc file.
- Please upload all figure files as individual ones, including the supplementary figure files.
- Please add your main, supplementary figure, and table legends to the main manuscript text after the references section.
- Please add the X and Bluesky handles of your host institute/organisation, as well as your own and/or one of the authors, in our system.
- Please reorder the title page so that it starts with the title, followed by the list of authors, and then their affiliations, etc.
- Please upload your Tables in editable .doc or Excel format; They can be included at the bottom of the main manuscript file or sent as separate files.
- The "Data Availability" section should be placed after the Materials & Methods section. Please consult our guidelines at <https://www.life-science-alliance.org/manuscript-prep#format>
- Please add a Conflict of Interest statement to your main manuscript text.
- Please upload your main figure 2 as a single file; figures will be displayed in-line in the HTML version of your paper, so please provide them as single-page files (Figures); we do not have a limit on the number of main figures, and these can be split if necessary for space.
- Please edit the call-out for fig. S3F, as there are no panels named as such.
- Please add callouts for Figures S1A-M; S6B-C; S7A-D; S8A-G and all videos to your main manuscript text.
- Please be sure that the authorship listing and order is correct

LSA now encourages authors to provide a 30-60 second video where the study is briefly explained. We will use these videos on social media to promote the published paper and the presenting author (for examples, see

<https://docs.google.com/document/d/1-UWCfbE4pGcDdcgzcmiuJl2XMBJnxKYeqRvLLrLSo8s/edit?usp=sharing>). Corresponding or first-authors are welcome to submit the video. Please submit only one video per manuscript. The video can be emailed to contact@life-science-alliance.org

A. FINAL FILES:

B. MANUSCRIPT ORGANIZATION AND FORMATTING:

Thank you for your attention to these final processing requirements. Please revise and format the manuscript and upload materials as soon as you are able.

Sincerely,

Sarita Hebbar, PhD
Scientific Editor
Life Science Alliance
<http://www.lsjournal.org>

Reviewer #1 (Comments to the Authors (Required)):

The revised manuscript successfully addresses the concerns and recommend suitable to be published.

Reviewer #2 (Comments to the Authors (Required)):

The authors have sufficiently addressed the technical issues I had raised, in particular they clarified that - although they created the transgenic reporter anew - it is based on a previously characterized plasmid. When viewed with appropriate caution, the provided data should indeed be a useful resource and I thus can now recommend publication.

Reviewer #3 (Comments to the Authors (Required)):

My concerns from the prior round of review were largely addressed.

-Kindly confirm if following images are different images:

(a) Figure 3B, WT and Figure S5A, WT

(b) Figure 3B, yap1 Crispant and Figure S5A, yap1 Crispant,

-> Yes, these are the same. Thanks for highlighting. We now include a sentence in the figure legend for S5A stating. "The two top row images, WT and yap1 Crispant also feature in Figure 3B and S3A." **Done**

(c) Figure S5A, WT and Figure S3 WT,

(d) Figure S5A, yap1 Crispant and Figure S3, yap1 Crispant.

If the same image has been utilised in multiple panels, then please indicate as such in the figure legends of all the panels.

-> Yes, these are the same. We now include a sentence in the figure legend for S3 and S3 stating. "The two top row images, WT and yap1 Crispant also feature in Figure 3B and S5A." **Done**

-Thank you for providing a detailed list of all used materials in the table named as 'Materials'. We encourage you to make a general reference to this table at the start of the methods section.

Done

-Please include details of objectives (Name, numerical aperture) and the temperature for live-imaging in the methods.

DONE Objective on SP5 was a Leica HC Plan Apo 20x NA 0.70 Ph2, and added temp.

-We encourage you to change the title of one of the methods sub-sections called 'Opera Imaging'. We suggest 'Whole body imaging for macrophage sub-types' as an option.

Thanks, now renamed as suggested **Done**

-Please provide the details for reagents and methods for RNA-FISH (Supplementary Figure 6).

DONE

-For Western Blot quantification, please clarify if loading values were determined by use of a loading control as the materials table mentioned the use of "Gapdh 1 Ab".

This sentence in the methods, "Loading levels were determined by analysing the peak height between ~55-100 kDa (where possible) of a Ponceau blot." refers to another approach for obtaining loading levels.

Please clarify this sentence and the use of Gapdh 1 Ab.

Ponceau was used for loading control as stated. Gapdh has been deleted from the table.

-Kindly include details of the secondary antibodies in the table.

DONE

-Please upload your main manuscript text as an editable doc file.

DONE

-Please upload all figure files as individual ones, including the supplementary figure files.

DONE

-Please add your main, supplementary figure, and table legends to the main manuscript text after the references section.

DONE

-Please add the X and Bluesky handles of your host institute/organisation, as well as your own and/or one of the authors, in our system.

DO

@gram-lab.bsky.social

@edinuni-irr.bsky.social

@fengfishlab.bsky.social

@michaelanosfairley.bsky.social

-Please reorder the title page so that it starts with the title, followed by the list of authors, and then their affiliations, etc.

DONE

-Please upload your Tables in editable .doc or Excel format; They can be included at the bottom of the main manuscript file or sent as separate files.

They are editable as is.

-The "Data Availability" section should be placed after the Materials & Methods section. Please consult our guidelines at <https://www.life-science-alliance.org/manuscript-prep#format>

DONE

-We have uploaded the RNAseq data and refer to the data deposited twice. "GSE311765".

-Please add a Conflict of Interest statement to your main manuscript text.

DONE

-Please upload your main figure 2 as a single file; figures will be displayed in-line in the HTML version of your paper, so please provide them as single-page files (Figures); we do

not have a limit on the number of main figures, and these can be split if necessary for space.

FIGURE PREPARED AS ONE PAGE. And new Figure 2 is uploaded

-Please edit the call-out for fig. S3F, as there are no panels named as such.

DONE

-Please add callouts for Figures S1A-M; S6B-C;S7A-D; S8A-G and all videos to your main manuscript text.

DONE

To clarify, in our manuscript we have:

Instead of specifically calling out S1A-M, we have three call outs for the general "Sup Fig 1"

Instead of specifically calling out S6B-C, we have one call out for the general "Sup Fig 6"

Instead of specifically calling out S7A-D, we have two call outs for the general "Sup Fig 7"

Instead of specifically calling out S8A-G, we have one call out for the general "Sup Fig 8"

In addition, we have now added call out twice for video 1-4.

-Please be sure that the authorship listing and order is correct

OK

- Upload: Graphical overview

DONE

We also included additional information in the acknowledgement section and edited a couple of typos. And have included the graphical overview, that was part of the initial submission.

December 3, 2025

RE: Life Science Alliance Manuscript #LSA-2025-03296-TRR

Dr. Carsten Gram Hansen
University of Edinburgh
Centre for Inflammation Research
Institute for Regeneration and Repair
Edinburgh BioQuarter, 4-5 Little France Drive
Edinburg EH16 4UU
United Kingdom

Dear Dr. Hansen,

Thank you for submitting your Research Article entitled "In vivo screen reveals specific roles of Hippo pathway components in development and regeneration". It is a pleasure to let you know that your manuscript is now accepted for publication in Life Science Alliance. Congratulations on this interesting work.

Your manuscript will now progress through copyediting and proofing. At the proofing stage, we remind you to mention the reuse of images for figures in all the figure legends (including Figure 3) as previously indicated.

It is journal policy that authors provide original data upon request.

DISTRIBUTION OF MATERIALS:

Again, congratulations on a very nice paper. I hope you found the review process to be constructive and are pleased with how the manuscript was handled editorially. We look forward to future exciting submissions from your lab.

Sincerely,

Sarita Hebbar, PhD
Scientific Editor
Life Science Alliance
<http://www.lsajournal.org>